# An Approach to Evaluate the Effective Cytoplasmic Concentration of Bioactive Agents Interacting with a Selected Intracellular Target Protein

**DOI:** 10.3390/pharmaceutics15020324

**Published:** 2023-01-18

**Authors:** Yuri V. Khramtsov, Alexey V. Ulasov, Andrey A. Rosenkranz, Tatiana A. Slastnikova, Tatiana N. Lupanova, Georgii P. Georgiev, Alexander S. Sobolev

**Affiliations:** 1Laboratory of Molecular Genetics of Intracellular Transport, Institute of Gene Biology of Russian Academy of Sciences, 34/5 Vavilov St., 119334 Moscow, Russia; 2Faculty of Biology, Lomonosov Moscow State University, 1-12 Leninskie Gory St., 119234 Moscow, Russia

**Keywords:** intracellular concentrations, Nrf2, Keap1, equilibrium model, CETSA, exogenous competitors, thermophoresis

## Abstract

To compare the effectiveness of various bioactive agents reversibly acting within a cell on a target intracellular macromolecule, it is necessary to assess effective cytoplasmic concentrations of the delivered bioactive agents. In this work, based on a simple equilibrium model and the cellular thermal shift assay (CETSA), an approach is proposed to assess effective concentrations of both a delivered bioactive agent and a target protein. This approach was tested by evaluating the average concentrations of nuclear factor erythroid 2-related factor 2 (Nrf2) and Kelch-like ECH-associated-protein 1 (Keap1) proteins in the cytoplasm for five different cell lines (Hepa1, MEF, RAW264.7, 3LL, and AML12) and comparing the results with known literature data. The proposed approach makes it possible to analyze both binary interactions and ternary competition systems; thus, it can have a wide application for the analysis of protein–protein or molecule–protein interactions in the cell. The concentrations of Nrf2 and Keap1 in the cell can be useful not only in analyzing the conditions for the activation of the Nrf2 system, but also for comparing the effectiveness of various drug delivery systems, where the delivered molecule is able to interact with Keap1.

## 1. Introduction

For the treatment of various diseases, it is often necessary for biologically active molecules to enter a cell and interact with certain target proteins. Some molecules themselves are able to penetrate intracellular barriers, while others, on the contrary, require some kind of drug delivery system. In any case, to evaluate the effectiveness of these biologically active molecules, the problem of their intracellular concentration determination often arises. In the simplest case, binary systems can be considered, namely, a biologically active molecule and a target protein. A more complex case is ternary competition systems, where the interaction of a biologically active molecule with a target protein leads to the release of another protein responsible for the biological effect. An example of this can be a well-known system for regulating gene expression, which is responsible for protecting cells from oxidative stress.

Oxidative stress is a common feature of numerous diseases including, but not limited to, neurodegenerative (Parkinson’s and Alzheimer’s diseases, amyotrophic lateral sclerosis, multiple sclerosis, Huntington’s disease [1,2]), inflammatory (atherosclerosis, rheumatoid arthritis, inflammatory bowel disease [3]), cardiovascular [4], and airway diseases [5]. On the one hand, the cells are regularly attacked by a diverse array of oxidants, originating from both endogenous metabolic systems and exogenous sources: UV light, ozone, cigarette smoke, and other toxicants, to name a few [6]. On the other hand, the reactive oxygen species are the well-known signaling molecules, playing an important role in various physiological processes [7,8]. In order to balance cellular oxidative status and to protect from the excess of oxidative molecules, our cells produce a broad set of enzymes, capable of neutralizing oxidants and resulting oxidation products [9]. Most of these cytoprotective genes are regulated by nuclear factor erythroid 2-related factor 2 (Nrf2) transcription factor, which is generally recognized as a master regulator of the cellular defense system [10,11,12]. Under normoxia, the Nrf2 interacts with its negative regulator, Kelch-like ECH-associated-protein 1 (Keap1), via low-affinity DLG and high-affinity ETGE motifs. Bound to the Keap1, the Nrf2 is inactive and constantly targeted to proteasomal degradation. The oxidative stress induces the modification of the multiple Keap1 cysteines, which leads to the conformation alteration, resulting in Nrf2 release from the complex with subsequent transport into the nucleus and activation of target gene expression [9,10]. In addition to protecting against oxidative stress, Nrf2 serves to protect against environmental pollutants, bacterial and viral toxins, and also has anti-inflammatory effects [13].

The Nrf2 system activation is of high pharmacological interest and is considered as a therapeutic strategy for different indications [5,14,15]. Specific competition for Keap1 between endogenous Nrf2 and exogenous peptides, based on the Nrf2 sequence, is one of the possible options [13,16,17,18,19]. Meanwhile, such a competitive approach should take into account intracellular concentrations of Nrf2 and Keap1. Moreover, knowledge of the concentrations of the components is necessary, not only in the quantitative analysis of the activation conditions of the Nrf2 system, but also in the general case when studying any intracellular protein–protein or molecule–protein interactions. In addition, one usually needs not just the average concentration of molecules in the cytoplasm, but its effective concentration. In other words, the concentration of only those molecules that are capable of interacting with the target protein, i.e., at least located with the target protein in the same cellular compartment, is usually required. In this article, based on the cellular thermal shift assay (CETSA) [20,21,22], we proposed a new approach to estimate the effective concentration of a molecule capable of interacting with a selected target protein, both in the case of binary and ternary competitive systems. This approach was tested by evaluating the concentrations of Nrf2 and Keap1 for several cell lines and comparing the results with known literature data.

## 2. Materials and Methods

### 2.1. Cell Lines

The murine macrophage RAW264.7, Lewis lung carcinoma (3LL, known also as LLC-1), hepatocellular carcinoma Hepa 1-6, and alpha mouse liver 12 (AML12) cell lines were maintained according to the specifications of the American Type Culture Collection (ATCC, Manassas, VA, USA). Mouse embryonic fibroblasts (MEF) were kindly provided by Dr. Moisenovich M. M., Faculty of Biology, Lomonosov Moscow State University, Moscow, Russia, and maintained according to the specifications of the ATCC.

### 2.2. Recombinant Proteins Used in the Work

To interact with the Keap1, we produced a fusion construct consisting of R1 monobody to the Keap1 protein [23] fused with a modular nanotransporter [24]. This fusion construct is abbreviated hereinafter as fmKeap1. In addition to the R1 monobody, this construct contains a ligand module capable of interacting with EGFR receptors on the cell surface, an endosomolytic module that allows it to exit endosomes, and a carrier module that combines all modules together and imparts additional solubility to the construct. Detailed characteristics of this construct will be published soon (the manuscript is in preparation).

To determine the affinity between the fmKeap1 and Keap1 proteins, the Keap1 C-terminal fragment (aa 312–624), responsible for the Nrf2 binding, hereinafter abbreviated as tKeap1, was used. tKeap1 was obtained from the Addgene plasmid #28025, encoding full size Keap1.

Expression of tKeap and fmKeap1 was carried out in *Escherichia coli* strain BL21(DE3). Cells were grown on LB Broth Miller—Novagen—with kanamycin (33 μg/mL) for tKeap and ampicillin (100 μg/mL) for fmKeap1. The tKeap was purified from inclusion bodies as described [25]. The fmKeap1 was purified from the soluble fraction as described [26]. Then purified tKeap and fmKeap1 were dialyzed against 10 mmol/L Na-phosphate buffer (pH 8.0) with 150 mmol/L NaCl.

### 2.3. Thermophoresis

The interaction affinity between fmKeap1 and tKeap1 were measured with a Monolith NT.115 instrument (NanoTemper Technologies, München, Germany) in phosphate buffer (25 mM NaH_2_PO_4_ (Sigma-Aldrich, Taufkirchen, Germany), 150 mM NaCl, pH 8.0). The tKeap1 protein was labeled with the Cy3 fluorescent dye. To do this, a 10-fold molar excess of the activated Cy3-N-hydroxysuccinimide ester (Lumiprobe, Moscow, Russia) was added to the tKeap1 protein in 65 mM carbonate buffer (pH 8.5), and the mixture was incubated for 1h at room temperature with constant stirring. The tKeap1 protein with attached Cy3 was separated from the free dye using PD10 chromatographic column. As a result, an average of 3.9 Cy3 molecules attached to one tKeap1 molecule. At a fixed concentration of tKeap1-Cy3 (40 nM), thermophoresis curves were obtained. Four such curves were obtained for each experiment, and the whole experiment was repeated three or four times. For each curve, the dissociation constant of the tKeap1 complex with a protein containing monobody to Keap1 was determined by Monolith NT.115 Instruments software (NanoTemper Technologies), it was averaged over all 12–16 curves, and the relative measurement error was determined.

### 2.4. Estimation of Cell Volumes

The characteristic sizes of the cells and their nuclei were measured with Leica TCS SP2 confocal microscope (Leica, Wetzlar, Germany) in a transmitted light channel (air lens with the magnification ×63 and NA 0.7 was used). For this purpose, the cells were detached from the substrate with a 0.25% trypsin solution, the medium was replaced with the Versene solution, and the cells were placed into a hemocytometer counting chamber, which is then placed in the microscope holder. The length, L, and width, W, of the cells and their nuclei were measured. To calculate the volume, V, the cells and their nuclei were approximated by a rotation ellipsoid, i.e., the calculation was carried out according to the formula:V = π∙L∙W^2^/6(1)

Volumes of cells and their nuclei obtained by expression (1) were averaged over 60–80 cells. Dividing cells were disregarded. The height was considered the same as the width. To verify this assumption, 3LL cells were stained with the CellMask^TM^ Green plasma membrane stain (Thermo Scientific, Waltham, MA, USA). Excitation was performed by the 532 nm laser, and fluorescence was recorded in the range 550–600 nm. Three-dimensional images of cells were obtained on the confocal microscope, and their length, width, and height were measured. We have shown that the height of the cells is the same as their width. The volume of the cytoplasm was considered as the difference between the volume of the cell and the volume of the nucleus, where the latter was determined by Equation (1).

The correctness of determination of the size of the nuclei according to the transmitted light channel was verified for the Hepa 1-6 cell line as follows. Nuclei were stained with the SYBR Green fluorescent dye (488 nm excitation, 500–550 nm fluorescence). The average volumes of the nuclei were determined from the data obtained from the transmitted light channel and the fluorescent channel 500–550 nm.

The correctness of the determination of the cells sizes according to microscopy was verified for 3LL line cells using the dynamic light scattering method on a ZetaPALS device (Brookhaven, GA, USA). To do this, one can find the diameter of the sphere, which has the same volume as calculated by expression (1) according to microscopy data and compare it with the hydrodynamic diameter determined from the data of dynamic light scattering. The dynamic light scattering measurement was carried out at the angle of 90° and the accumulation time of 30 s in seven replications.

### 2.5. Cellular Thermal Shift Assay (CETSA)

The cellular thermal shift assay (CETSA) [21] was conducted as follows. Cells were harvested via trypsin treatment and centrifugation (200× *g*, 5 min). Then, cells were suspended in a buffer (pH 8.0) containing 25 mM NaH_2_PO_4_, 150 mM NaCl, 1.5 μg/mL aprotinin, 0.174 mg/mL phenylmethylsulfonyl fluoride, and 5 mM EDTA (cells from 25 cm^2^ cultural flask). The cells were quantified using a MACSQuant Analyzer flow cytometer (Miltenyi Biotec GmbH, Paris, France). Melting curves of the studied proteins were obtained both in intact cells and in cell lysates. In the first case, the cell suspensions were heated to set temperatures in the range of 40–50 °C for 3 min, while one probe remained unheated. Then, the cells were cooled to room temperature and lysed by four freezing–thawing cycles. Freezing was performed in liquid nitrogen. In the second case, the cells were first lysed by four freezing–thawing cycles, and then the cell lysates were heated to set temperatures in the range of 40–50 °C for 3 min. Then, for both settings, the cell lysates were centrifuged (8000× *g*, 60 min, 4 °C). The supernatants of the cell lysates were applied to denaturing electrophoresis followed by Western blot staining by either anti-Nrf2 (ab31163, Abcam, Cambridge, UK) or anti-Keap1 (ab139729, Abcam) mAb. These antibodies have been successfully used to assess the relative amounts of Nrf2 and Keap1 using Western blot on a number of murine cell types [27,28,29,30,31,32]. Sample electrophoresis was performed using standard 10% SDS-PAGE. Given the frequent inhomogeneous coloration of samples obtained using Western blot, all samples were loaded on the gel in triplicate. This enables excluding random outliers of individual bands from the analysis and obtaining more reliable averaged data. Samples were transferred from the gel to a supported nitrocellulose membrane (0.22 μm) using the Trans-Blot Turbo Transfer System (Bio-RAD, Hercules, CA, USA). For each sample, the band intensity in the selected area and the background area (Appendix A) were measured, and they were subtracted from each other. For Keap1, a band of about 70 kDa was used (Appendix A), which corresponds to the literature data for Keap1 [20,31,32]. For Nrf2, in most of the cell lines studied by us, two bands are observed—about 64 kDa and 28 kDa (Appendix A), and, for further calculations, a band of about 64 kDa was used. It should be noted that for the 3LL line, the band of approximately 64 kDa disappeared (Appendix A). Therefore, a different cocktail of inhibitors was used for this cell line, namely, Halt Protease Inhibitor Single-Use Cocktail (100×) (Thermo Scientific, Waltham, MA, USA). In this case, a ~64 kDa band appeared again (Appendix A). After incubation of 3LL cells for 2 h with 20 μM proteasome inhibitor MG-132 (Selleckchem.com), another band of about 85 kDa appeared on the Western blot (Appendix A). For each studied cell line, the obtained band intensity was averaged for three identical samples and normalized to the band intensity of the sample not subjected to heating treatment for CETSA. The resulting curve was interpolated by sigmoid function in the Origin 6.0 software. For each cell line and staining with the selected antibodies, Western blots were performed in *n* ≥ 4 replicates.

### 2.6. Model Description

#### 2.6.1. Binary Interactions

For a quantitative analysis of the interaction of a molecule delivered to the cytoplasm, M, with a target protein, T, the following equilibrium model was proposed. Hereinafter, only the effective concentrations of the components are considered, i.e., only those molecules that are in the same cellular compartment and are able to interact with each other. The following designations were introduced in the presented model: [M_tot_]—the total concentration of the studied molecule in the cytoplasm; [T_tot_]—the total concentration of the target protein in the cytoplasm; [M]—the concentration of free molecule in the cytoplasm; [T]—the concentration of free target protein in the cytoplasm; [M•T]—the concentration of the M•T complex in the cytoplasm; K_d_—the dissociation constant of the M•T complex.

Assuming a 1:1 binding stoichiometry, we obtain the following equation:(2)M+T⇔M•T

Dissociation constant, K_d_, at equilibrium:(3)$$K_{d}=\frac{\left[M\right]\cdot [T]}{[M•T]}$$

From Equation (2) and considerations of material balance:(4)$${[M}_{\mathrm{tot}}]=[M]+[M•T];\;{[T}_{\mathrm{tot}}]=[T]+[M•T]$$

From Equation (3) and system of Equation (4):(5)$${[M}_{\mathrm{tot}}]=[M]+\frac{\left[M\right]\cdot \left[T\right]}{K_{d}};\;{[T}_{\mathrm{tot}}]=[T]+\frac{\left[M\right]\cdot \left[T\right]}{K_{d}}$$

By solving the system of Equation (5) one can obtain:(6)$${[M}_{\mathrm{tot}}]=\frac{T_{\mathrm{tot}}-T}{T}\cdot \left(K_{d}+T\right)$$
or
(7)$${[M}_{\mathrm{tot}}]=\frac{1-F\left(T\right)}{F\left(T\right)}\cdot \left(K_{d}+F\left(T\right)\cdot [T_{\mathrm{tot}}]\right)$$
where F(T) denotes the ratio:F(T) = [T]/[T_tot_](8)

In the cell lysate at a sufficiently high dilution, when the condition
F(T)·[T_tot_] << K_d_ or [T_tot_] << K_d_/F(T)(9)
is met, Equation (7) changes to:(10)$${[M}_{\mathrm{tot}}]\approx \frac{1-F\left(T\right)}{F\left(T\right)}\cdot K_{d}$$

The accuracy of [M_tot_] estimation according to Equation (10) depends on how strongly inequality (9) is satisfied. So, if [T_tot_] is ten times less than K_d_/F(T), then Equation (10) gives an underestimation of [M_tot_] by ten percent.

#### 2.6.2. Ternary Competition Systems

To assess the cytoplasmic concentration of molecules, M, capable of interacting with a target protein, T, and displacing a protein, C, from the complex with T, the following model was proposed. For simplicity, the total cell volume minus the nucleus volume was taken as the cytoplasm volume, similarly to Iso et al. [33]. The following designations were introduced in the presented model: [C_tot_]—the total concentration of C in the cytoplasm; [T_tot_]—the total concentration of T in the cytoplasm; [M_tot_]—the total concentration of M in the cytoplasm; [T]—the concentration of free T in the cytoplasm; [C]—the concentration of free C in the cytoplasm; [M]—the concentration of free M in the cytoplasm; [C•T]—the concentration of the C•T complex in the cytoplasm; [M•T]—the concentration of the M•T complex in the cytoplasm; K_d1_—the dissociation constant of the C•T complex; K_d2_—the dissociation constant of the M•T complex.

The model considers the time when the system is in the equilibrium. It is also assumed that all proteins bind in 1:1 stoichiometry.

Thus, consider the following system of interactions:(11)C+T⇔C•T;M+T⇔M•T

For equilibrium dissociation constants, the following equations are satisfied:(12)$$K_{d1}=\frac{\left[C\right]\cdot [T]}{[C•T]};\;K_{d2}=\frac{\left[M\right]\cdot [T]}{[M•T]}$$

From the system of interactions (11) and considerations of material balance:(13)$${[C}_{\mathrm{tot}}]=[C]+[C•T];\;{[M}_{\mathrm{tot}}]=[M]+[M•T];\;{[T}_{\mathrm{tot}}]=[T]+[C•T]+[M•T]$$

From Equation (12) and system of Equation (13):(14)$${[C}_{\mathrm{tot}}]=[C]+\frac{\left[C\right]\cdot [T]}{K_{d1}};\;{[M}_{\mathrm{tot}}]=[M]+\frac{\left[M\right]\cdot \left[T\right]}{K_{d2}};\;{[T}_{\mathrm{tot}}]=[T]+\frac{\left[C\right]\cdot \left[T\right]}{K_{d1}}+\frac{\left[M\right]\cdot \left[T\right]}{K_{d2}}$$

By solving the system of Equation (14) one can obtain:(15)$${[M}_{\mathrm{tot}}]=\left(\frac{{[T}_{\mathrm{tot}}]}{\frac{{[C}_{\mathrm{tot}}]}{\left[C\right]}-1}-K_{d1}-[C]\right)\cdot \frac{K_{d1}\cdot \left(\frac{{[C}_{\mathrm{tot}}]}{\left[C\right]}-1\right)+K_{d2}}{K_{d1}}$$

In the case when [M_tot_] = 0, Equation (15) will turn into:(16)$$\frac{{[T}_{\mathrm{tot}}]}{\frac{{[C}_{\mathrm{tot}}]}{\left[C\right]}-1}=K_{d1}+[C]$$

If we denote
F(C) = [C]/[C_tot_](17)

Then,
(18)$$[T_{\mathrm{tot}}]=\frac{1-F\left(C\right)}{F\left(C\right)}\cdot \left(K_{d1}+F\left(C\right)\cdot \left[C_{\mathrm{tot}}\right]\right)$$

Which is the same as Equation (7), only with a different designation of the parameters.

#### 2.6.3. Nrf2 System

When describing the interaction of M molecules with the Keap1•Nrf2 complex, one can use the model proposed in the previous paragraph. In this case, the T protein will be the Keap1 protein, and the C protein will be the Nrf2 protein. The model is also considered to be close to equilibrium and it does not take into account either the rate of change in the concentration of molecules, capable of interacting with the Keap1, or the subsequent transport of Nrf2 into the nucleus with activation of Nrf2 expression [34]. The assumption that all proteins bind in 1:1 stoichiometry is valid, because two binding sites of Keap1 differ greatly in affinity to Nrf2 (the dissociation constants of 5 nM and 1 μM [35]). Therefore, we took into account only the binding site with high affinity. With the new designation of the parameters, Equation (15) will look like:(19)$${[M}_{\mathrm{tot}}]=\left(\frac{{[\mathrm{Keap}1}_{\mathrm{tot}}]}{\frac{{[\mathrm{Nrf}2}_{\mathrm{tot}}]}{\left[\mathrm{Nrf}2\right]}-1}-K_{d1}-[\mathrm{Nrf}2]\right)\cdot \frac{K_{d1}\cdot \left(\frac{{[\mathrm{Nrf}2}_{\mathrm{tot}}]}{\left[\mathrm{Nrf}2\right]}-1\right)+K_{d2}}{K_{d1}}$$

Additionally, Equation (16) will look like:(20)$$\frac{{[\mathrm{Keap}1}_{\mathrm{tot}}]}{\frac{{[\mathrm{Nrf}2}_{\mathrm{tot}}]}{\left[\mathrm{Nrf}2\right]}-1}=K_{d1}+[\mathrm{Nrf}2]$$

If we denote
F(Nrf2) = [Nrf2]/[Nrf2_tot_](21)

Then,
(22)$$\frac{{[\mathrm{Keap}1}_{\mathrm{tot}}]}{\frac{1}{F(\mathrm{Nrf}2)}-1}=K_{d1}+F(\mathrm{Nrf}2)\cdot [{\mathrm{Nrf}2}_{\mathrm{tot}}]$$

Equation (22) can be rewritten as:(23)$$[{\mathrm{Nrf}2}_{\mathrm{tot}}]=\frac{{[\mathrm{Keap}1}_{\mathrm{tot}}]}{1-F\left(\mathrm{Nrf}2\right)}-\frac{K_{d1}}{F\left(\mathrm{Nrf}2\right)}$$

Or
(24)$$[{\mathrm{Keap}1}_{\mathrm{tot}}]=\left(1-F\left(\mathrm{Nrf}2\right)\right)\cdot \left(\left[{\mathrm{Nrf}2}_{\mathrm{tot}}\right]+\frac{K_{d1}}{F\left(\mathrm{Nrf}2\right)}\right)$$

In the case when
[Nrf2_tot_] << K_d1_/F(25)

(In the cell lysate at a sufficiently high dilution), dependence (24) will turn into:(26)$$[{\mathrm{Keap}1}_{\mathrm{tot}}]\approx K_{d1}\cdot \frac{1-F\left(\mathrm{Nrf}2\right)}{F\left(\mathrm{Nrf}2\right)}$$

Which is a special case of expression (10). On average, for the samples we studied, [Nrf2_tot_] was sixteen times less than K_d1_/F (fulfillment of condition (25)), i.e., Equation (26) gives an underestimation of [Keap1_tot_] by six percent.

Similarly, according to Equation (10) and denoting:F(Keap1) = [Keap1]/[Keap1_tot_](27)

We obtain
(28)$$[{\mathrm{Nrf}2}_{\mathrm{tot}}]\approx K_{d1}\cdot \frac{1-F(\mathrm{Keap}1)}{F(\mathrm{Keap}1)}$$

In a cell it should be taken into account that the addition of molecules capable of interacting with Keap1 will lead to a change in the total concentration of Nrf2. The following additional designations have been introduced into the model: [Nrf2_tot0_]—the total concentration of Nrf2 in the cytoplasm without the addition of molecules capable of interacting with Keap1; *k*_1_—degradation rate constant of bound Nrf2; *k*_2_—degradation rate constant of free Nrf2. Before the addition of molecules capable of interacting with Keap1, the rates of synthesis and degradation of Nrf2 molecules in the cytosol are equal *k*_1_·[Nrf2_tot0_], since almost all of Nrf2 is in complex with Keap1. Upon addition of molecules capable of interacting with Keap1, the degradation rates of Nrf2 molecules will change and become equal *k*_1_·[Nrf2•Keap1] + *k*_2_·[Nrf2]. The rates of synthesis and degradation are equal at equilibrium, so
(29)$$k_{1}\cdot [\mathrm{Nrf}2•\mathrm{Keap}1]+k_{2}\cdot [\mathrm{Nrf}2]=k_{1}\cdot [{\mathrm{Nrf}2}_{\mathrm{tot}0}]$$

Taking into account that [Nrf2_tot_] = [Nrf2•Keap1] + [Nrf2], from Equation (29), one obtains:(30)$$\frac{\left[{\mathrm{Nrf}2}_{\mathrm{tot}}\right]}{\left[{\mathrm{Nrf}2}_{\mathrm{tot}0}\right]}=\frac{1}{1-F\left(\mathrm{Nrf}2\right)\cdot \left(1-\frac{k_{2}}{k_{1}}\right)}$$

Substituting Equation (30) into Equation (19), one obtains:(31)$${[M}_{\mathrm{tot}}]=\left(\frac{F\left(\mathrm{Nrf}2\right)}{1-F\left(\mathrm{Nrf}2\right)}\cdot {[\mathrm{Keap}1}_{\mathrm{tot}}]-K_{d1}-\frac{F\left(\mathrm{Nrf}2\right)\cdot [{\mathrm{Nrf}2}_{\mathrm{tot}0}]}{1-F\left(\mathrm{Nrf}2\right)\cdot \left(1-\frac{k_{2}}{k_{1}}\right)}\right)\cdot \frac{K_{d1}\cdot \frac{1-F\left(\mathrm{Nrf}2\right)}{F\left(\mathrm{Nrf}2\right)}+K_{d2}}{K_{d1}}$$

All presented dependencies were plotted and processed in Origin 6.0 software.

## 3. Results

To assess the effective cytoplasmic concentration of a bioactive agent, we proposed the following scheme (Figure 1). Cell concentration was determined using flow cytometry. The volumes of the cells and their nuclei were measured using microscopy. For Hepa 1-6 cell line, the average nucleus volumes according to the transmitted light channel and the fluorescent channel were 410 ± 20 and 444 ± 19 μm^3^, respectively. Thus, according to the data of the transmitted light channel, it is possible to correctly determine the volumes of the nuclei. The cell size determined by the dynamic light scattering is in good agreement with the size measured by the microscopy. For example, the average diameter of the 3LL line cells determined by the microscopy was 13.9 ± 0.4 μm, and the diameter determined by the dynamic light scattering was 13.2 ± 2.7 μm. The volume of the cytoplasm was taken as the difference between the volume of the cell and its nucleus. Knowing the concentration of cells and the volume of the cytoplasm, it is possible to calculate the dilution factor of the studied molecule during cell lysis.

The approach proposed by us was applied for a quantitative description of the Nrf2 system. To estimate the concentration of a molecule in our approach, it is necessary to verify the correctness of the proposed mathematical description of the interactions between the studied molecules. To test the assumptions underlying Equation (19) regarding the interaction stoichiometry and the equilibrium dissociation constant value, K_d1_, we used concentrations of half-maximal inhibition of the Keap1 with Nrf2 interaction (IC_50_) [16] and the equilibrium dissociation constants measured [16] using surface plasmon resonance for a number of peptides interacting with Keap1 via ETGE motif (Table 1). Knowing [Keap1_tot_] = 100 nM [36] and K_d1_ = 5 nM [35], from Equation (19) assuming that at IC_50_ [Nrf2]/[Nrf2_tot_] = 0.5 and [Nrf2] = 50 nM, we can theoretically calculate K_d2_ (Table 1) from the known IC_50_. For most used peptides, this evaluation shows a good agreement between the data obtained theoretically and the known experimental data on K_d2_ (Table 1), which verifies the assumptions underlying Equation (19).

For a quantitative analysis of a selected cell line, we used the cellular thermal shift assay, CETSA [21]. This assay enables us to obtain the so-called melting curve of a studied protein both in intact cells and in cell lysates; moreover, this curve depends on whether this protein forms a complex with another molecule or not [21]. In the present work, these curves were obtained for the Nrf2 protein according to Western blot data. We chose the 64 kDa band to obtain the Nrf2 melting curve, since among all the cell lines studied by us, the band above 64 kDa is observed only for the MEF cell line, and, according to the literature data, the 57–68 kDa band corresponds to the Nrf2 fragment, and its intensity changes upon activation of the Nrf2 system [37,38,39]. Another band observed at 28 kDa also seems to correspond to the Nrf2 fragment [40]. Moreover, the melting curves obtained from the 64 and 28 kDa bands coincide with each other (Appendix A). When the proteasome inhibitor MG-132, which causes a significant increase in the Nrf2 concentration, was added to 3LL cells, a band of ~85 kDa appeared on the blot, which was also visible on lysates of MEF cells (Appendix A). The melting curves obtained from the ~85 kDa band coincide with similar curves obtained from the ~64 kDa band (Appendix A). When in a cell, the Nrf2 system was not initially activated; then, almost all Nrf2 protein molecules are in complex with Keap1 protein molecules [19]. The initially inactive state of the Nrf2 system is typical for almost all cells, with the exception of some cancer cells. Figure 2a shows the melting curve of such Keap1-associated Nrf2 averaged over three cell lines (Hepa1, MEF, and RAW264.7). A melting curve for free Nrf2 can be obtained by adding to the cell lysate an excess of a protein that can interact with the Keap1 protein and lead to the displacement of the Nrf2 protein from the complex with Keap1 protein. The fmKeap1 protein was used for this purpose. To test the ability of the fmKeap1 to interact with the Keap1 protein, the thermophoresis method was used (Figure 3). The dissociation constant of the complex of the fmKeap1 with the tKeap1 protein determined by this method was 7.9 ± 3.3 nM. The addition of an excess of this construct to the cell lysate will lead to the complete release of Nrf2 from the complex with Keap1 and will allow one to obtain a free Nrf2 melting curve. The same dependence takes place for Nrf2 in lysates of different cell lines. Figure 2a shows such a curve for free Nrf2 averaged over the studied cell lines.

In some cancer cell lines, the Nrf2 system may already be activated [41]. Based on the Nrf2 melting curve, this can be assumed for the 3LL cell line (Figure 2a). For this cell line, the melting curve of Nrf2 in the cell is shifted significantly towards the melting curve characteristic for free Nrf2 (Figure 2a). In contrast, the melting curve of Nrf2 for the randomly selected non-tumor cell line AML12 coincides with the melting curve of bound Nrf2 (Figure 2a).

Using the melting curves of Nrf2, we can use our proposed method to determine the proportion of free Nrf2 in a cell or cell lysate, F(Nrf2) (Equation (21)). For this, at a fixed temperature T_i_, the difference of ordinates for the curves corresponding to the studied sample and Nrf2 in complex with Keap1 Δ_i_(T_i_) is obtained (Figure 2a). At the same temperature, the difference in modulus of the relative intensities of the bands for the curves corresponding to free Nrf2 and Nrf2 in complex with Keap1, Δ_max_(T_i_) is determined (Figure 2a). Then the fraction of free Nrf2 at the chosen temperature T_i_, F(T_i_), will be equal to F(T_i_) = Δ_i_(T_i_)/Δ_max_(T_i_). To determine F(Nrf2) at physiological temperatures, one can extrapolate the dependence F = F(T_i_) linearly to a temperature of 37 °C (Figure 2b). For extrapolation, the linear part of the curve starting at 42 °C is used.

Previously, Iso et al. [33] estimated the number of Nrf2 and Keap1 molecules for several cell lines. However, for quantitative analysis of Nrf2-system, it is necessary to know not only the total number of molecules per cell, but also the concentration of these molecules. For this purpose, the volumes of the cells and their nuclei were measured (Table 2). The literature data [33] showed that the concentration of Nrf2 in the cytoplasm is close to the average concentration of Nrf2 in the cell, while Keap1 is mainly located in the cytoplasm and not in the nucleus. Therefore, the concentration of Nrf2 in the cytoplasm was estimated by dividing the known amount of Nrf2 in the cell [33] by the cell volume (Table 2). The concentration of Keap1 in the cytoplasm was obtained by dividing the known amount of Keap1 in the cell [33] by the difference in the cell and nucleus volumes (Table 2).

Using our approach, we determined the F(Nrf2) parameter for the lysate of selected cell lines and, using Equation (26), determined the average concentration of Keap1 in this lysate. Knowing the volume of cells and their nuclei, as well as the number of cells in 1 μL determined by flow cytometry before their lysis, one can calculate the average concentration of Keap1 in the cytoplasm. The right column of Table 2 shows the cytoplasmic concentrations of Keap1 obtained by this way. It should be noted that all the obtained concentrations of Keap1 are in good agreement with the concentrations of Keap1 calculated on the basis of the literature data in the cytoplasm of cells (Table 2), which confirms the applicability of our proposed approach and indicates that the melting curve for the Nrf2 fragment selected for analysis (64 kDa) coincides with the melting curve for the full-sized Nrf2. Moreover, for the four cell lines studied (Table 2, except for the 3LL cell line), in which the Nrf2 system was initially not activated (the melting curves of Nrf2 coincide with the melting curve of Nrf2 in complex with Keap1, Figure 2a), the concentration of Keap1 in the cytoplasm is the same within the standard errors of experiments and averaged 269 ± 4 nM.

For the 3LL cell line, using our approach, we can determine the F (proportion of free Nrf2) parameter in intact cells (Figure 2a,b). Further, using Equation (23) and a certain average concentration of Keap1 in the cytoplasm, one can estimate the average concentration of Nrf2 in the cell. It turned out that the Nrf2 concentration obtained by this way, within the experimental errors, coincides with the Nrf2 concentration calculated on the basis of the literature data on the amount of Nrf2 in 3LL cells (Table 2). For other cell lines, where almost all of Nrf2 is in the complex with Keap1, the F parameter in intact cells is close to zero; therefore, Equation (23) cannot be used to estimate the average concentration of Nrf2 in the cell.

According to Equation (28), the average concentration of Nrf2 in a cell can be determined by our approach, using Western blot with anti-Keap antibodies to determine F(Keap1) (Equation (27)). To do this, one needs to obtain the melting curves of free and bound Keap1. A curve for bound Keap1 can be obtained by performing CETSA in intact cells (Figure 4a). For the studied cell lines, these curves coincide, and the averaged curve is presented in Figure 4a (blue curve). However, it is not possible to obtain a curve for free Keap1 due to the unknown number of different endogenous Nrf2 competitors for Keap1 binding. Nevertheless, for the 3LL cell line, this curve can be reconstructed using the following considerations. Knowing the concentration of cells, the volume of the 3LL cell, and the volume of its nucleus, as well as the previously determined average concentration of Nrf2, it is possible to calculate F(Keap1) at physiological temperatures in triplicate curves of the obtained cell lysate samples. Assuming a linear dependence of F(Keap1) on temperature in the range of 37–48 °C, as was observed for F in the range of 42–48 °C (Figure 2b), and also assuming that no melting of free Keap1 is observed at 42 °C, we can estimate the values of F(Keap1) at different temperatures in the range of 42–48 °C for each of the cell lysate samples. Then, using the melting curves of Keap1 in the cell lysate and in the intact cells, one can obtain the expected melting curve of free Keap1. Such a curve averaged over several repetitions is presented in Figure 4a (red curve). Using this curve, it is possible to determine F(Keap1, T_i_) = Δ_i_(T_i_)/Δ_max_(T_i_) for any cell line at different temperatures (same as Figure 2a) and then extrapolate linearly to a temperature of 37 °C (Figure 4b). The obtained F(Keap1) values for cell lysates, as well as cell volumes and their initial concentration, were used to estimate the average concentration of Nrf2 in the cell (Table 2, second column from the right).

It should be noted that the estimated average concentrations of Nrf2 in the cell, obtained by the proposed method, agree quite well with the concentrations of Nrf2 calculated based on the literature data (Table 2), which indicates the correctness of the proposed melting curve for free Keap1. This makes it possible to estimate the concentration of Nrf2 in any randomly selected cell line. This was conducted for the AML12 cell line (Table 2). It was shown that for the four cell lines studied (Table 2, except for the 3LL cell line), in which almost all of Nrf2 is in the complex with Keap1, the average concentration of Nrf2 in the cell was similar and averaged 62 ± 5 nM.

For a randomly selected cell line in which the Nrf2 system is not initially activated, it can be assumed that the concentrations of Nrf2 and Keap1 are 63 and 269 nM, respectively. Then, there are two approaches to estimate the concentration of a molecule that competes with Nrf2 for binding to Keap1. The protein proposed by us containing a monobody to Keap1 can act as such a molecule. In the first approach, the concentration of this protein can be estimated from the Keap1 melting curves and expression (7). In the second approach, the concentration of this protein can be estimated from the Nrf2 melting curves and Equation (31). Figure 5 shows the dependence of the estimated protein concentration on the proportion of protein-bound Keap1 (Figure 5a) or free Nrf2 (Figure 5b) in the cell. In this case, the *k*_2_/*k*_1_ ratio was taken equal to 10 based on the difference in the rate of degradation of free Nrf2 and Nrf2 in the complex with Keap1 [42]. It can be seen that the estimated range of protein concentrations is significantly wider in the case of the analysis for the Nrf2 protein (Figure 5b) than in the case of the analysis for the Keap1 protein (Figure 5a). Usually, in the proposed approach, the error in determining the proportion of protein-bound Keap1 or free Nrf2 was about 0.05. Thus, in the analysis of binary interaction and ternary competition systems, the intracellular concentration of the studied fmKeap1 with an error less than 30% can be determined in the ranges of 40–310 and 190–1100 nM, respectively. This demonstrates that both options complement each other well, and their joint use greatly expands the area of practical application of the proposed approach.

## 4. Discussion

In the proposed approach aimed at estimation of the concentration of selected molecules, it is necessary, first of all, to determine the number of cells and the sizes of cells and their nuclei, which make it possible to calculate the dilution factor of molecules during cell lysis. Then, one needs to determine the proportion of target protein molecules bound to the selected molecules in the cell lysate, followed by determination of the concentration of the molecules in the cell lysate, based on a simple mathematical description of the interaction between the molecule and the target protein and subsequent calculation of the concentration of the molecules in the cytoplasm using the dilution factor (Figure 1). Cell concentration can be measured using a hemocytometer counting chamber, as is usually performed in many laboratories, but we used flow cytometry as a more accurate method. The characteristic dimensions of cells can be measured by various methods convenient for a particular laboratory. We measured the size of cells detached from the substrate using optical microscopy. The obtained dimensions determined by optical microscopy and dynamic light scattering are in good agreement with each other. Optical microscopy allows one to determine the size of not only the cells, but also their nuclei. However, the size of the nuclei should only be determined if the chosen molecule is known to reside predominantly in the cytoplasm rather than in the nucleus. If this is not known, then it is sufficient to estimate the average concentration of the molecule in the cell.

The main advantage of the proposed approach is that it can be used to estimate not just the concentration of selected molecules, but their effective concentration, i.e., the concentration of only those molecules that are able to interact with the selected target protein. In a cell, the effective concentration of molecules can differ markedly from their average concentration due to, for example, the fact that some of the selected molecules can undergo modifications that block binding to the target protein or due to the fact that the molecule and the target protein can be located in different cellular compartments. Evaluation of the effective concentration is achieved due to the fact that the behavior of the target protein, rather than the molecule itself, is studied. In other words, the proportion of target protein molecules bound to the studied molecules is estimated, and the effective concentrations of these molecules are calculated using a mathematical interaction model. The model can take into account whether this is a simple two-component interaction or competitive binding with some third component. The proportion of target protein molecules bound to the studied molecules was determined in the proposed approach using CETSA. In world practice, this method is quite common for studying the interaction of proteins with their ligands [22,43], and we have shown that it can also be used to determine the desired proportion of the target protein.

The proposed approach was tested on the Nrf2 system. On the one hand, this system is interesting in itself, due to its responsibility for protecting cells from oxidative stress and for detoxification. On the other hand, it has already been studied in many respects, which makes it possible to use known data to verify the proposed approach. Then, considering Keap1 as the target protein, one can determine the concentration of Nrf2, and, if one chooses Nrf2 as the target protein, one can determine the concentration of Keap1. First of all, it was necessary to determine the mathematical model of interactions in this system. In the general case, for the ternary system, where the delivered molecule competes with the Nrf2 protein for binding to the Keap1 protein, an equilibrium model was chosen, where the searchable concentration of the molecule is found by expression (19). It is assumed that one Nrf2 molecule or one competing molecule binds to Keap1. This assumption is based on the fact that the two Keap1 binding motifs in the Nrf2 molecule strongly differ in the affinity of the interaction, allowing us to consider only a high-affinity interaction based on the ETGE motif [35]. It is also known that the dissociation constant of the Keap1:Nrf2 complex is 5 nM [35]. To test the assumptions about the stoichiometry of interactions, as well as the dissociation constants of the Keap1:Nrf2 complex, we used published data on the interaction of a number of ETGE motif containing peptides with Keap1 leading to Nrf2 displacement [16]. It turned out that for most of these peptides, their interaction with the Nrf2-Keap1 system is well described by Equation (19) (Table 1). Thus, Equation (19) correctly describes the competitive interaction observed in the Nrf2 system, which means that Equations (26) and (28), which are special cases of Equation (19), will also describe the system quite well.

In the proposed approach, to estimate the concentration of molecules, it is necessary to obtain a melting curve of the target protein using some selected band from a Western blot. Moreover, these curves are normalized (for a sample that is not subject to heating), so their appearance does not change when the percentage of this band in the sample changes. For Keap1, band selection is easy, because, based on the literature data, full-length Keap1 on electrophoresis has a band of about 70 kDa [20,31,32]. The situation with Nrf2 is much more complicated. The fact is that there are conflicting opinions in the literature on this issue. On the one hand, given that Nrf2 has a molecular weight of 60–70 kDa, in many articles the band at 57–68 kDa is taken for Nrf2 [37,38]. Moreover, it was shown that this band is indeed related to Nrf2, since it changes in response to activation of the Nrf2 system [39]. However, other studies have shown that this band corresponds rather to the Nrf2 fragment, while full-length Nrf2, due to its abnormal electrophoretic mobility, presents as a band at 95–110 kDa [37,38]. However, obtaining such a band in a cell lysate is complicated by the fact that there are various Nrf2 isoforms in the cell, which can differ significantly in molecular weight [38], and also by the fact that an unknown part of Nrf2 in the cell undergoes various modifications and partial cleavage, which can occur both during cell lysis and in the cell itself and not only in proteasomes [40,44]. For analysis, we chose a band of about 64 kDa. A different cocktail of protease inhibitors was chosen to make this band appear in the 3LL cell line Western blot (Appendix A). On the Western blot of the studied cell lines, a band of about 28 kDa is also clearly visible (Appendix A). Moreover, the melting curves obtained from the bands of either about 64 kDa or about 28 kDa coincide with each other (Appendix A); therefore, to estimate the concentration using the proposed approach, the band of about 28 kDa can also be used. Assuming that the 64 kDa band corresponds to the Nrf2 fragment, an attempt was made to obtain a band corresponding to the full-length Nrf2. For this, 3LL cells were incubated with the known proteasome inhibitor MG-132. Due to the inhibition of one of the main pathways of Nrf2 degradation, this inhibitor leads to a significant increase in the total concentration of Nrf2 [45]. Even with the use of this inhibitor, the band at 95–110 kDa was not observed; however, an increase in the concentration of Nrf2 results in the appearance of a band of about 85 kDa, close in molecular weight to the full-sized Nrf2 (Appendix A). Taking into account the unknown molecular weight of Nrf2 isoforms and their possible modifications for the 3LL line, the 85 kDa band can be considered as corresponding to Nrf2, which is close to the full-size form. For the MEF and 3LL cell lines, we have shown that the melting curves obtained from the band of about 85 kDa coincide with the curves obtained from the band of about 64 kDa (Appendix A). Thus, the band of about 64 kDa that we have chosen makes it possible to obtain a melting curve corresponding to the melting curve of the full-size Nrf2. This is also confirmed by the fact that the Keap1 concentrations obtained from the Nrf2 melting curves using the 64 kDa band are in good agreement with the literature data (Table 2).

Normally, a concentration of free Nrf2 in cytoplasm is low, as it is predominantly in the complex with Keap1 repressor protein, which leads to Nrf2 proteasomal degradation. Under oxidative stress, the Nrf2 is released from the complex with the Keap1, its concentration increases, it enters the cell nucleus, and, activating expression of a number of genes, triggers cell defense processes. In some, predominantly cancerous, cell lines, the proportion of free Nrf2 in the cytosol can be significant; in other words, the Nrf2 system in these cells is initially activated [41]. We have shown that the CETSA makes it possible to determine in which cells all Nrf2 is in a complex with Keap1, and in which the proportion of free Nrf2 is significant, i.e., in which cells the Nrf2 system is initially activated and in which it is not. To do this, it is first necessary to obtain melting curves for free Nrf2 and Nrf2 in complex with Keap1. The second curve can be generated using CETSA for most cell lines where all of Nrf2 is complexed with Keap1. This, for example, was conducted for Hepa1, MEF, and RAW264.7 cell lines (Figure 2a). The curve for free Nrf2 was obtained in the cell lysate, to which an excess of protein containing anti-Keap1 monobody was added. The preservation of the ability of such a chimeric protein to bind to Keap1 was tested using the thermophoresis method. The melting curve for free Nrf2 is also independent of the cell type used. Thus, to determine whether the Nrf2 system is activated or not in an arbitrary cell line, it is necessary to obtain the Nrf2 melting curve in this cell line using CETSA and then compare it with the melting curves of free Nrf2 and Nrf2 in complex with Keap1. If it strongly differs from the melting curve for Nrf2 in complex with Keap1 towards the melting curve for free Nrf2, then the Nrf2 system in this cell line is probably activated. This is observed, for example, for the 3LL cell line used in our work (Figure 2a). In contrast, for the AML12 cell line, the Nrf2 system is not initially activated (Figure 2a).

Previously, attempts were made to quantitatively describe the Nrf2 system [19,46]. Despite the rather detailed description of all possible interactions in this system [19,46], for its practical application, there is not enough knowledge about the exact average concentrations of Nrf2 and Keap1 in the cell. Although Iso et al. [33] determined the amounts of Nrf2 and Keap1 per cell for a number of cell lines, the concentration was calculated only for RAW264.7 cell line. In our work, we first took four cell lines used in the work of Iso et al. [33] and, using optical microscopy, determined the sizes of cells and their nuclei for them (Table 2). For RAW264.7 cell line, Iso et al. [33] published a different cell volume [47] than that obtained by us. However, they relied on the Coulter counter data [47], where the result is highly dependent on a calibration. Thus, our data on the sizes of cells and their nuclei appear to be more accurate. These data, as well as Nrf2 and Keap1 amounts determined by Iso et al. [33], made it possible to calculate the concentrations of Nrf2 and Keap1 in four cell lines (Table 2).

Using the melting curves for free Nrf2 and Nrf2 in a complex with Keap1 and the melting curve for Nrf2 in the test sample, we can use our approach to determine the proportion of free Nrf2 in a cell or cell lysate, F(Nrf2) (expression (21)). This, in turn, can be used to estimate the average concentration of Keap1 in the cytoplasm according to expression (26). To do this, it is necessary to recalculate the concentration of Keap1 obtained in the cell lysate into the concentration in the cytoplasm using the calculated dilution factor. The good agreement between the obtained values of the Keap1 concentration and those previously calculated by us (Table 2) proves the efficiency of our proposed approach. In addition, for the 3LL line, expression (23) can be used to determine the average concentration of Nrf2 in the cell. It also agrees well with that obtained earlier (Table 2).

If anti-Keap1 antibodies rather than anti-Nrf2 antibodies are used in CETSA, then it is possible to determine the proportion of free Keap1 in the cytoplasm, F(Keap1), (expression (27)) and from it to determine the average concentration of Nrf2 in the cell using expression (28). However, there is no melting curve for free Keap1 for this. Normally, Keap1 interacts in the cell with both Nrf2 and various endogenous Nrf2 competitors for binding to Keap1 [48,49]. Adding excess protein capable of interacting with Keap1 to the cell lysate will result in a melting curve of Keap1 in complex with this protein, rather than a melting curve of free Keap1. To overcome this difficulty, we reconstructed the melting curve of free Keap1. For this, F(Keap1) was calculated for the 3LL cell line from a known concentration of total Nrf2 at physiological temperatures. Next, we assumed a linear dependence of F(Keap1) on temperature, and also, that at 42 °C, no melting of free Keap1 was observed. The latter assumption was made on the basis of the literature data that the Keap1 complex with a small molecular weight inhibitor did not melt at a temperature of 42 °C [50]. The assumptions made it possible to reconstruct the melting curve for free Keap1 (Figure 4a, red curve) and determine F(Keap1) and, hence, the average concentration of Nrf2 for all studied cell lines. It should be noted that the obtained values of the Nrf2 concentration are in good agreement with the values previously obtained by us, which proves the correctness of the reconstruction of the free Keap1 curve.

Thus, the proposed approach makes it possible to determine the average concentrations of Nrf2 and Keap1 in the cytoplasm for any selected cell line. Mouse hepatocyte AML12 cells were chosen as such a cell line, which is interesting in terms of protecting the liver from oxidative stress. As shown in Figure 2a, this cell line does not show initial activation of the Nrf2 system. Comparison of all studied cell lines according to the average concentration of Nrf2 and Keap1 in the cytoplasm showed that for all cells where the initial activation of the Nrf2 system is not observed, the concentrations of Nrf2 or Keap1 do not depend on the type of the selected cell line and are 62 ± 5 and 269 ± 4 nM, respectively (Table 2). In other words, all differences in Nrf2 or Keap1 quantities per cell observed by Iso et al. [33] can be attributed to the differences in cell sizes. Considering the strong difference between the types of selected cells, it can be assumed that for any randomly selected cell line, the concentrations of Nrf2 and Keap1 can be close to the aforementioned concentrations. These values can be further used as parameters in expression (19).

When assessing the concentration of any molecule capable of interacting with Keap1 and displacing Nrf2 at the same time, it should be borne in mind that during such an interaction, the total concentration of Nrf2 in the cell will change [19,46]. Therefore, the equilibrium model proposed by us requires some extension, becoming more similar to model 1 from [46]. Namely, this model introduces the degradation rates of bound and free Nrf2, *k*_1_ and *k*_2_. In this case, expression (19) turns into expression (31). To plot the dependence of the concentration of the studied molecule on the fraction of free Nrf2, it is only necessary to estimate the ratio *k*_2_/*k*_1_ in expression (31). This can be obtained from the literature data on the ratio of the lifetimes of free and bound Nrf2, t_1_/t_2_. One can consider this ratio close to 10 [42].

In this work, the Nrf2 system was considered as a test system on which the proposed approach was checked. However, the approach itself can be very useful for further study of the activation conditions for this system. To determine the required concentrations of molecules, we considered only the first stages of the Nrf2 system activation. Namely, the release of Nrf2 from the complex with Keap1 in response to external action and an increase in the concentration of Nrf2 due to the difference in the rates of degradation of free Nrf2 and Nrf2 in the complex with Keap1. All other stages of the process of activation of the Nrf2 system, such as the entry of Nrf2 into the cell nucleus, the activation of the corresponding genes, and the production of protective cell proteins, are not considered in this approach. Therefore, the proposed approach does not cancel a large number of methods used to study the activation of the Nrf2 system, but only complements them in terms of Nrf2 and Keap1 concentrations. Moreover, the assessment of these concentrations does not depend on the cause underlying the release of Nrf2 from the complex with Keap1. This can be either the competition of a certain molecule and Nrf2 for binding to Keap1, as this is considered in our work, or, for example, the release of Nrf2 from the complex with Keap1 due to the modification of Keap1 sensor groups. In any case, the developed approach makes it possible to evaluate how the concentrations of Nrf2 and Keap1 change under some influence on the cell and, thus, to better understand the conditions for the activation of the Nrf2 system. As a rule, only changes in the relative amounts of Nrf2 and Keap1 in the cell are studied using Western blot, while there are few studies that determine the absolute amounts of these proteins. In the latter case, as a rule, calibration dependencies with known amounts of the recombinant protein are used. This is how the quantities of Nrf2 and Keap1 per cell were obtained [33], which were used by us to verify the proposed approach. Indeed, in all cases, it turned out that the concentrations of Nrf2 and Keap1, estimated using our approach, are in good agreement with these literature data.

The proposed approach makes it possible not only to study the onset of activation of the Nrf2 system, but also to use this system to compare the effectiveness of different drug delivery systems. Indeed, let us consider an arbitrary delivery system for an effector molecule capable of interacting with Keap1 and leading to the release of Nrf2. A number of approaches, such as a Western blot with appropriate protein calibrations, allow estimation of the concentration of this delivery system in the cytoplasm. However, in a number of cases, the delivery system enters the cell, but not the cytosol, being trapped in closed membrane formations—endosomes—with only some proportion entering finally the cytosol. In other words, the concentration of the delivery system in the cytoplasm may differ significantly from the concentration of this system in the cytosol. As already mentioned, the approach proposed by us makes it possible to determine not just the average concentration of a molecule, but its effective concentration, i.e., the concentration of only those molecules that are able to interact with the target protein. Taking into account that sometimes the effector molecule can be cleaved from the delivery system in the cell, in our example this will mean that only the concentration of effector molecules released from endosomes into the cytosol and retaining the ability to bind to Keap1 will be determined. Thus, for the chosen delivery system, it is possible to evaluate its effectiveness in delivering the effector molecule directly to the target protein, in the case under consideration, to Keap1. This, in turn, makes it possible to study the influence of the structure of the delivery system on the efficiency of delivery and, in the long run, to significantly improve this efficiency. In the proposed approach, we have described a mathematical model of both a simple two-component interaction (Equation (7)) and a ternary, competitive interaction (Equation (15)). Experimentally, in the first case, cell lysis is performed first and then CETSA and Western blot with antibodies to Keap1; in the second case, it is first CETSA and then cell lysis and Western blot with antibodies to Nrf2. These two options complement each other well for the Nrf2/Keap1 system. The first of them makes it possible to analyze the onset of a drug delivery system entry into cells when its concentration is still low (40–310 nM for fmKeap1). On the contrary, the second variant makes it possible to study the late stages of a drug delivery system entry into the cell, when its concentrations can be rather high (190–1100 nM for fmKeap1). We are going to explore the possibility of using this approach to modulate the antioxidant response in target cells using modular nanotransporter technology. This technology makes it possible to deliver drug cargoes to a specified compartment of a target cell [51,52]. According to our unpublished data, a comparison of the effective concentration of fmKeap1 (one of the modular nanotransporters) in the cytoplasm, obtained using our proposed approach, with the concentration of fmKeap1 obtained from Western blot data with antibodies to fmKeap1 and the corresponding calibrations for the fmKeap1 concentration, showed that these two approaches, within experimental errors, give the same concentration fmKeap1 (article is in preparation). This, on the one hand, once again confirms our approach, and, on the other hand, suggests that the contribution of the modular nanotransporters remaining in endosomes to their total concentration in the cytoplasm is not large, apparently due to their further degradation in lysosomes.

It should be noted that the proposed approach is not limited to the described models, where the concentration of the studied molecule is determined by Equations (7) and (15). The stoichiometry of the interaction may differ from 1:1, and the specific mathematical expression for the concentration of the molecule may be different. However, the proposed approach makes it possible to determine the fraction of a free soluble target protein in the cell cytoplasm and, based on it, to calculate the concentration of a molecule capable of interacting with a specified target protein. To do this, it is only necessary to obtain melting curves for a free target protein, a target protein in complex with a given molecule, and a target protein in cells containing the molecule under study. Then, it is possible to obtain the proportion of the free target protein in the studied cell sample at different temperatures and extrapolate it to physiological temperatures. It should be noted that the studied molecule can be both exogenous and endogenous. It is only important to determine in some way the constant of its interaction with the target protein.

The proposed approach can be applied not only to the analysis of Western blot data. Thus, CETSA can be set up much easier and faster with fluorescently labeled molecules, as has been performed, for example, for Keap1 labeled with green fluorescent protein [20] or SARS-CoV-2 virus nucleocapsid protein labeled with mRuby3 fluorescent protein [53]. Other options to set up the CETSA method are possible. The proposed approach makes it possible to determine the concentrations of interacting molecules without calibrations, which in some cases cannot be obtained. This is the case, for example, when the target protein cannot be isolated in its pure form. Or, on the contrary, for low molecular weight unlabeled molecules, in the case when it is impossible to directly determine their intracellular concentration. Moreover, the indisputable advantage of the method is that it determines the concentration of molecules that can interact with the target protein inside the cell, in other words, have already overcome all intracellular barriers. As already discussed, this will help to improve the efficiency of various drug delivery systems. In other words, the proposed approach potentially has a very wide practical application.

## 5. Conclusions

A new approach was proposed for assessing the effective cytoplasmic concentration of a molecule capable of interacting inside the cell with a selected soluble target protein. Moreover, variants of both direct interaction of the molecule with the target protein and competition for binding with another protein are possible. The concentrations of both exogenous and endogenous molecules can be estimated by this method. For several cell lines, the method was tested on the Nrf2 system and the total average concentrations of Nrf2 and Keap1 proteins in the cytoplasm were estimated, and they coincide with the known literature data on the amounts of these proteins in these cells. The proposed approach uses the melting curves of the selected target protein obtained using Western blot. The main advantage of the proposed approach is that it is not the total protein concentration in the cytoplasm that is obtained, but its effective concentration, i.e., only those protein molecules that are able to interact with the selected target protein are taken into account. Thus, the proposed approach can significantly complement the existing approaches for the analysis of protein–protein interactions in the cell. The concentrations of Nrf2 and Keap1 in the cell determined using this approach will not only make it possible to better study the conditions for the activation of the Nrf2 system in the cell, but also to use this system to compare the effectiveness of different drug delivery systems. This will better identify the key parameters that affect the efficiency of these delivery systems and further optimize their properties.

## Figures and Tables

**Figure 1 pharmaceutics-15-00324-f001:**
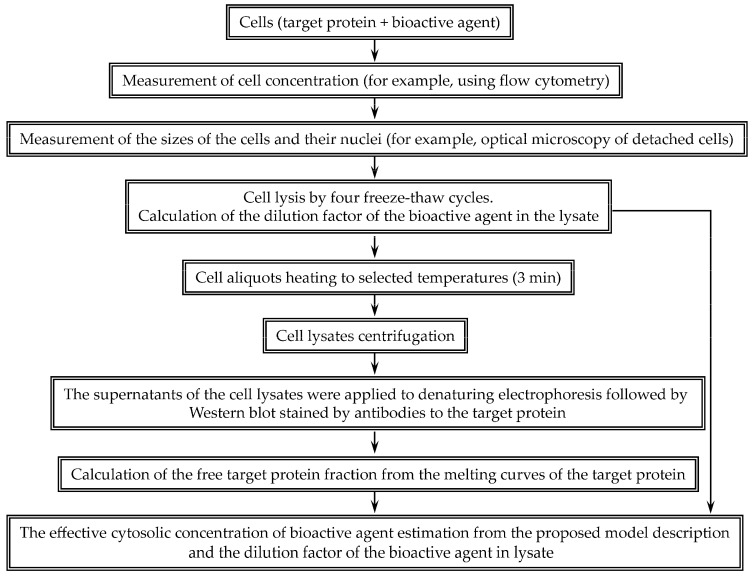
Scheme of our approach to evaluate the effective cytosolic concentration of bioactive agents interacting with a selected intracellular target protein.

**Figure 2 pharmaceutics-15-00324-f002:**
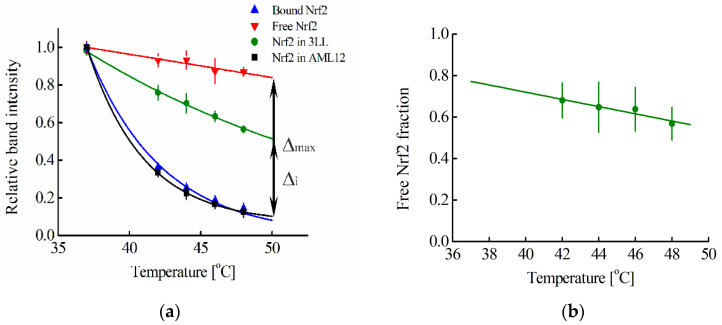
(**a**) Melting curves averaged over Hepa1, MEF, and RAW264.7 cell lines of free Nrf2 and Nrf2 bound to Keap1 and total Nrf2 in 3LL and AML12 cell lines. The dependences are normalized to the band intensity at 37 °C; (**b**) Dependence between free Nrf2 fraction (Δ_i_/Δ_max_) in 3LL cells and temperature. The green line shows the linear extrapolation of experimental data to 37 °C. Standard errors (SE) are shown (*n* = 4–12).

**Figure 3 pharmaceutics-15-00324-f003:**
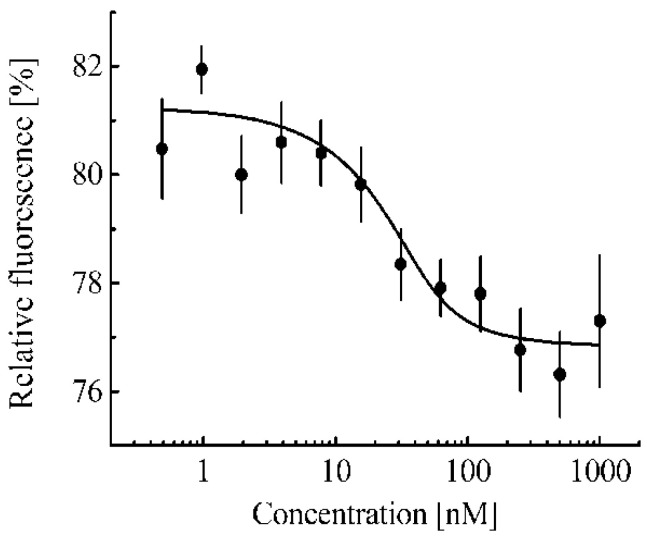
Dependences of relative fluorescence intensities (fluorescence intensity before the start of thermophoresis is taken as 100%) at 20 s after the start of thermophoresis on the concentration of the fmKeap1 protein at a constant concentration of the tKeap1 (40 nM). Standard errors (SE) of relative fluorescence intensities are shown (*n* = 8–12).

**Figure 4 pharmaceutics-15-00324-f004:**
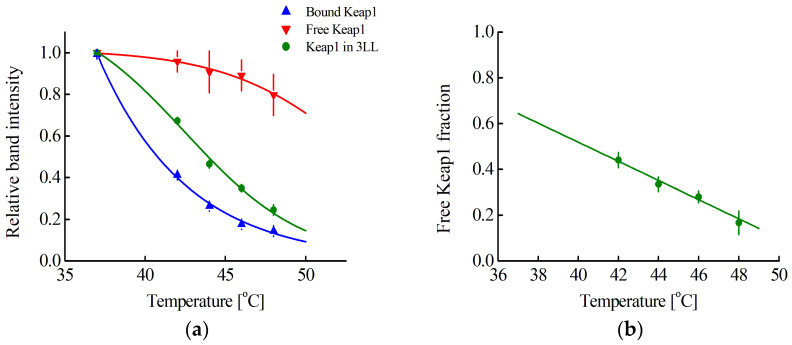
(**a**) Melting curves averaged over Hepa1, MEF, and RAW264.7 cell lines of bound Keap1 and Keap1 in 3LL cell lysate and reconstructed curve of free Keap1. The dependences are normalized to the band intensity at 37 °C; (**b**) Dependence between free Keap1 fraction in 3LL cell lysate and temperature. The green line shows the linear extrapolation of experimental data to 37 °C. Standard errors (SE) are shown (*n* = 4–12).

**Figure 5 pharmaceutics-15-00324-f005:**
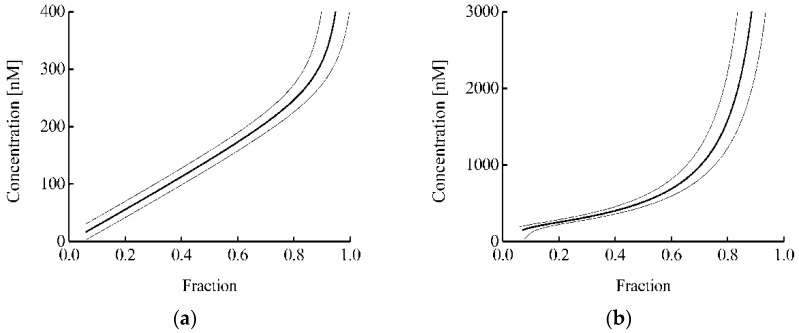
Dependences of the fmKeap1 protein concentration on the fraction of Keap1 bound to this protein (**a**) and free Nrf2 (**b**). Thin lines mark the range of standard errors.

**Table 1 pharmaceutics-15-00324-t001:** The inhibition of the Keap1:Nrf2 interaction by Nrf2 peptides.

Peptide Sequence	IC_50_, nM ^a^	K_d2_, nM Experimental ^a^	K_d2_, nM Theoretical ^b^
H-LDEETGEFL-OH	3480 ± 920	352	382 ± 102
H-LDEETGEFL-NH_2_	3570 ± 2200	355	392 ± 245
Ac-LDEETGEFL-OH	194 ± 49	21.1	16.6 ± 5.5
Ac-LDEETGEFL-NH_2_	196 ± 32	21.4	16.8 ± 3.6
H-QLDEETGEFL-OH	272 ± 26	27.3	25.2 ± 2.9
H-LQLDEETGEFL-OH	298 ± 33	31.3	28.1 ± 3.7
H-QLQLDEETGEFL-OH	249 ± 22	23.8	22.7 ± 2.4
H-FAQLQLDEETGEFL-OH	243 ± 20	22.5	22.0 ± 2.2
H-AFFAQLQLDEETGEFL-OH	163 ± 11	23.9	13.1 ± 1.2

^a^ [16]; ^b^ calculated from Equation (19). The standard error values (±SE) are represented.

**Table 2 pharmaceutics-15-00324-t002:** Parameters estimated in this work.

Cell Lines	Cell Volume μm^3^	Nucleus Volume μm^3^	[Nrf2_tot_], nM	[Keap1_tot_], nM	[Nrf2_tot_], CETSA nM ^a^	[Keap1_tot_], CETSA nM ^b^
RAW264.7	1360 ± 70	222 ± 18	56 ± 4	256 ± 85	52 ± 16	276 ± 26
Hepa-1	1600 ± 50	410 ± 20	78 ± 4	276 ± 48	77 ± 17	259 ± 44
3LL	1410 ± 40	174 ± 10	227 ± 17	67 ± 22	259 ± 95 * 220 ± 21	69 ± 15
MEF	2270 ± 100	400 ± 20	61 ± 3	260 ± 119	59 ± 14	265 ± 38
AML12	1400 ± 40	200 ± 10	-	-	59 ± 16	274 ± 22

^a^ Obtained by the CETSA with antibodies to the Keap1 protein and Equation (28). ^b^ Obtained by the CETSA with antibodies to the Nrf2 protein and Equation (26). * Obtained by the CETSA with antibodies to the Nrf2 protein and Equation (23). The standard error values (±SE) are represented (in the last two columns *n* = 4–6).

## Data Availability

Not applicable.

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
