# Peer review of "An Approach to Evaluate the Effective Cytoplasmic Concentration of Bioactive Agents Interacting with a Selected Intracellular Target Protein"

_pharmaceutics, 2023, doi:10.3390/pharmaceutics15020324_

Round 1

Reviewer 1 Report

An Approach to Evaluate the Effective Cytosolic Concentration of Bioactive Agents Interacting with a Selected Intracellular Target Protein 

This study is trying to evaluate cytoplasmic concentration of NRF2 and KEAP1 proteins within a cell. However, unfortunately there are critical issues in this manuscript. In Figure S1, Western blotting data with anti-NRF2 antibody shows a 25 kDa of band, which is inconsistent with the size of NRF2 reported from the other papers that show approximately 110 kDa of band for NRF2 protein. In addition, anti-NRF2 antibody (ab31163) used in this study is no longer guaranteed by the supplier (https://www.abcam.com/nrf2-antibody-ab31163.html). As for the anti-KEAP1 antibody (ab139729) used in this study, mouse Keap1 is not included that reacts with by this antibody (https://www.abcam.com/keap1-antibody-ab139729.html). Because this study heavily depends on the Western blotting, these unconvinced data are critical defect.

Author Response

“In addition, anti-NRF2 antibody (ab31163) used in this study is no longer guaranteed by the supplier (https://www.abcam.com/nrf2-antibody-ab31163.html). As for the anti-KEAP1 antibody (ab139729) used in this study, mouse Keap1 is not included that reacts with by this antibody (https://www.abcam.com/keap1-antibody-ab139729.html). Because this study heavily depends on the Western blotting, these unconvinced data are critical defect.”

Our Reply:

Thanks for your fair comments. Although the supplier of the anti-NRF2 antibody (ab31163) does not currently guarantee its suitability for Western blotting, the manufacturer's website does state that it has been used successfully by some users for this purpose. Moreover, in the same place on the site there are links to relevant articles, including those for mouse cells. For the KEAP1 antibody (ab139729), the manufacturer's website does not really indicate that they can be used to detect Keap1 in mouse lines. However, there are a number of works, links to which are available on the manufacturer's website, where these antibodies have been successfully used in both mouse and rat lines.

Relevant information with appropriate references has been added to Materials and Methods:

These antibodies have been successfully used to assess the relative amounts of Nrf2 and Keap1 using Western blot on a number of murine cell types [27, 28, 29, 30, 31, 32].

“In Figure S1, Western blotting data with anti-NRF2 antibody shows a 25 kDa of band, which is inconsistent with the size of NRF2 reported from the other papers that show approximately 110 kDa of band for NRF2 protein.”

Our Reply:

Thanks for the fair remark about the band  for Nrf2.  We actually made a mistake and presented a non-representative Western blot for the 3LL cell line, because in all other studied cell lines there was a band of about 64 kDa, based on which the calculations were made. We corrected this error by providing Western blots for all studied cell lines (new Figure S2). We performed additional experiments with another set of inhibitors, and for the 3LL cell line, a band of about 64 kDa also became visible. Moreover, in the proposed approach, only the melting curves Nrf2 and Keap1 are used. These curves are normalized (for a sample that is not subject to heating), so their appearance does not change when the percentage of the band in the sample changes.  And we have shown that the melting curves obtained from the bands of 28 and 64 kDa coincide with each other (new Figure S3). Moreover, to study whether a band closer to 95-110 kDa appears, we performed additional experiments on the incubation of 3LL cells with the proteasome inhibitor MG-132. Bands at 95-110 kDa were not observed, however, a band at 80-90 kDa appeared (new Figure S2). Taking into account the fact that there are different isoforms of Nrf2, which differ markedly in molecular weight, the band obtained is close to the band of the full-sized Nrf2. And for the MEF and 3LL cell lines, the bands around 85 and around 64 kDa give similar melting curves (new Figure S3). Thus, within the framework of our approach, the use of bands around 28, 64, and 85 kDa gives the same result, which corresponds to the melting curve of the full-sized Nrf2. This is also confirmed by the results obtained by us using the proposed approach. Using our approach, it is possible to obtain the concentration of Keap1 in the cytoplasm from the band for Nrf2. On the other hand, the concentration of Keap1 can be obtained from the literature Western blot data with appropriate calibrations for the concentration of Keap1. For the four cell lines, the concentrations obtained by the two different approaches are the same. Moreover, the Nrf2 concentration in the 3LL cell line was estimated from the Nrf2 melting curve. It also matched well with that calculated from the literature data. This confirms not only our approach, but also the fact that the melting curve used corresponds to Nrf2.

Phrases were inserted into Materials and Methods:

For Keap1, a band of about 70 kDa was used (Figure S1), which corresponds to the literature data for Keap1 [20, 32, 31]. For Nrf2, in most of the cell lines studied by us, two bands are observed - about 64 kDa and 28 kDa (Figure S2) and for further calculations, a band of about 64 kDa was used. It should be noted that for the 3LL line, the band of approximately 64 kDa disappeared (Figure S2e). Therefore, a different cocktail of inhibitors was used for this cell line, namely: Halt Protease Inhibitor Single-Use Cocktail (100x) (Thermo Scientific, USA). In this case, a ~ 64 kDa band appeared again (Figure S2f). After incubation of 3LL cells for 2 hours with 20 μM MG-132 (Selleckchem.com), another band of about 85 kDa appeared on the Western blot (Figure S2g).”

Phrases inserted into Results:

Moreover, the melting curve can be obtained both from the band of about 64 kDa and from the band of about 28 kDa and, importantly, these curves coincide with each other (Figure S3). When the proteasome inhibitor MG-132 is added to 3LL cells, a band of ~ 85 kDa appeared on the blot, which is also visible on lysates of MEF cells (Figure S2). The melting curves obtained from the ~ 85 kDa band coincide with similar curves obtained from the ~ 64 kDa band (Figure S3).”

Phrases inserted into Discussion:

In the proposed approach, to estimate the concentration of molecules, it is necessary to obtain a melting curve of the target protein using some selected band from a Western blot. Moreover, these curves are normalized (for a sample that is not subject to heating), so their appearance does not change when the percentage of this band in the sample changes. For Keap1, band selection is easy, because based on the literature data; full-length Keap1 on electrophoresis has a band of about 70 kDa [20, 31, 32]. The situation with Nrf2 is much more complicated. The fact is that there are conflicting opinions in the literature on this issue. On the one hand, given that Nrf2 has a molecular weight of 60-70 kDa, in many articles the band at 57-68 kDa is taken for Nrf2 [41, 42]. Moreover, it was shown that this band is indeed related to Nrf2, since it changes in response to activation of the Nrf2 system [43]. However, other studies have shown that this band corresponds rather to the Nrf2 fragment, while full-length Nrf2 due to its abnormal electrophoretic mobility presents as a band at 95-110 kDa [41, 42]. ]. However, obtaining such a band in a cell lysate is complicated by the fact that there are various Nrf2 isoforms in the cell, which can differ significantly in molecular weight [42], and also by the fact that an unknown part of Nrf2 in the cell undergoes various modifications and partial cleavage, which can occur both during cell lysis and in the cell itself and not only in proteasomes [46, 47]. For analysis, we chose a band about 64 kDa. A different cocktail of protease inhibitors was chosen to make this band appear in the 3LL cell line Western blot (Figure S2f). On the Western blot of the studied cell lines, a band of about 28 kDa is also clearly visible (Figure S2). Moreover, the melting curves obtained from the bands of either about 64 kDa or about 28 kDa coincide with each other (Figure S3), therefore, to estimate the concentration using the proposed approach, the band of about 28 kDa can also be used. Assuming that the 64 kDa band corresponds to the Nrf2 fragment, an attempt was made to obtain a band corresponding to the full-length Nrf2. For this, 3LL cells were incubated with the known proteasome inhibitor MG-132. Due to the inhibition of one of the main pathways of Nrf2 degradation, this inhibitor leads to a significant increase in the total concentration of Nrf2 [48]. But even with the use of this inhibitor, the band at 95-110 kDa was not observed; however, a band of about 85 kDa, close in molecular weight to the full-sized Nrf2, appeared (Figure S3g). Taking into account the unknown molecular weight of Nrf2 isoforms and their possible modifications for the 3LL line, the 85 kDa band can be considered as corresponding to Nrf2, which is close to the full-size form. For the MEF and 3LL cell lines, we have shown that the melting curves obtained from the band of about 85 kDa coincide with the curves obtained from the band of about 64 kDa (Figure S3c and S3f). Thus, the band of about 64 kDa that we have chosen makes it possible to obtain a melting curve corresponding to the melting curve of the full-size Nrf2.”

In Supplements, we have swapped Figures S1 and S2 and significantly expanded Figure S2. In addition, we have added Figure S3:

“Figure S3: Melting curves of Nrf2 for RAW264.7 (a), Hepa-1 (b), MEF (c), AML12 (d), 3LL (e) and 3LL with MG-132 (f) cell lysate obtained from a band of about 65 kDa (red curve), about 28 kDa (blue curve) and about 85 kDa (green curve). SE are shown (n=4).”

Reviewer 2 Report

Conclusion part is very short. It should focus more on the application or future aspects of the present research work.

Author Response

“Conclusion part is very short. It should focus more on the application or future aspects of the present research work.”  

Our Reply:

Thanks for your fair comment and suggestion. We have substantially expanded the conclusion to include possible future applications of our approach and emphasized the importance of determining the effective concentration of molecules.

New Conclusion:

A new approach was proposed for assessing the effective cytoplasmic concentration of a molecule capable of interacting inside the cell with a selected soluble target protein. Moreover, variants of both direct interaction of the molecule with the target protein and competition for binding with another protein are possible. The concentrations of both exogenous and endogenous molecules can be estimated by this method. For several cell lines, the method was tested on the Nrf2 system and the total average concentrations of Nrf2 and Keap1 proteins in the cytoplasm were estimated and they coincide with the known literature data on the amounts of these proteins in these cells. The proposed approach uses the melting curves of the selected target protein obtained using Western blot. The main advantage of the proposed approach is that it is not the total protein concentration in the cytoplasm that is obtained, but its effective concentration, i.e. only those protein molecules that are able to interact with the selected target protein are taken into account. Thus, the proposed approach can significantly complement the existing approaches for the analysis of protein-protein interactions in the cell. The concentrations of Nrf2 and Keap1 in the cell determined using this approach will not only make it possible to better study the conditions for the activation of the Nrf2 system in the cell, but also to use this system to compare the effectiveness of different drug delivery systems. This will better identify the key parameters that affect the efficiency of these delivery systems and further optimize their properties.”

Reviewer 3 Report

In the submitted manuscript, Khramtsov et al. have described an approach for evaluating intracellular concentrations of bioactive agents and their target proteins. This work seeks to ease the evaluation of the efficiency and effectiveness of drug delivery in complex model systems. The authors have thoroughly detailed their approach and have provided adequate and appropriate references which served as the foundation for this study. The model they have put forth could potentially aid in the estimation of critical intracellular drug/target concentrations, facilitating pharmacological advancements. However, a few limitations need to be addressed.

The main concern is the overuse of assumptions derived from scientific literature in assessing the validity of the models, instead of providing empirical evidence. The authors need to use quantitative proteomics to determine intracellular target protein abundance to ensure their model(s) are accurate. At a minimum, utilization of quantitative western blots with standard curves are needed. Validation of the model with a separate molecule and target outside of Keap1-Nrf2 would also increase confidence in the model.

Additionally, in the model description, lines 174 and 213, is there a cutoff value for defining '<<' or is this subjective?

For Figure 1a, the legend should be revised to clarify that the melting curves are averaged from Hepa1, MEF and RAW264.7 cells. It is misleading as it is currently written.

Author Response

“The main concern is the overuse of assumptions derived from scientific literature in assessing the validity of the models, instead of providing empirical evidence. The authors need to use quantitative proteomics to determine intracellular target protein abundance to ensure their model(s) are accurate. At a minimum, utilization of quantitative western blots with standard curves are needed. Validation of the model with a separate molecule and target outside of Keap1-Nrf2 would also increase confidence in the model.”

Our Reply:

Thanks for your fair comments and suggestions. In the text of the manuscript, we added an explanation that earlier in the literature, using the Western blot and the corresponding calibration dependences, the amounts of Nrf2 and Keap1 in some cell lines were estimated. It is these data that we used to verify our proposed approach. Regarding other molecules, not Nrf2 and Keap1, we are developing modular nanotransporters (MNT) – a drug delivery system. One of these MNTs is aimed at activating the NRF2 system and is mentioned in the manuscript as fmKeap1. An article is currently being prepared in which, among other things, a comparison of the approach proposed in this manuscript and Western blot with the necessary concentration calibrations will be given. We already have data that these two approaches give the same result, but it is supposed to be given in another article. In the text of the manuscript, we inserted a brief mention of this as an article that is being prepared for publication.

Phrases inserted into Discussion:

“In any case, the developed approach makes it possible to evaluate how the concentrations of Nrf2 and Keap1 change under some influence on the cell, and thus, to better understand the conditions for the activation of the Nrf2 system. As a rule, only changes in the relative amounts of Nrf2 and Keap1 in the cell are studied using Western blot, while there are few studies that determine the absolute amounts of these proteins. In the latter case, as a rule, calibration dependencies with known amounts of the recombinant protein are used. This is how the quantities of Nrf2 and Keap1 per cell were obtained [33], which were used by us to verify the proposed approach. Indeed, in all cases, it turned out that the concentrations of Nrf2 and Keap1, estimated using our approach, are in good agreement with these literature data.”

And

According to our unpublished data, a comparison of the effective concentration of fmKeap1 (one of the modular nanotransporters) in the cytoplasm, obtained using our proposed approach, with the concentration of fmKeap1 obtained from Western blot data with antibodies to fmKeap1 and the corresponding calibrations for the fmKeap1 concentration showed that these two approaches, within experimental errors, give the same concentration fmKeap1 (article is in preparation). This, on the one hand, once again confirms our approach, and, on the other hand, suggests that the contribution of the modular nanotransporters remaining in endosomes to their total concentration in the cytoplasm is not large, apparently due to their further degradation in lysosomes.

“Additionally, in the model description, lines 174 and 213, is there a cutoff value for defining '<<' or is this subjective?”

Our Reply:

Thanks for your question.

Phrases were inserted into Materials and Methods:

The accuracy of [Mtot] estimation according to equation (10) depends on how strongly inequality (9) is satisfied. So, if [Ttot] is ten times less than Kd/F(T), then equation (10) gives an underestimation of [Mtot] by ten percent.

And

On average, for the samples we studied, [Nrf2tot] was sixteen times less than Kd1/F (fulfillment of condition (25)), i.e., the equation (26) gives an underestimation of [Keap1tot] by six percent.

“For Figure 1a, the legend should be revised to clarify that the melting curves are averaged from Hepa1, MEF and RAW264.7 cells. It is misleading as it is currently written.”

Our Reply:

Thank you, we have added this information:

Figure 2. (a) – Melting curves averaged over Hepa1, MEF and RAW264.7 cell lines of free Nrf2 and Nrf2 bound to Keap1 and total Nrf2 in 3LL and AML12 cell lines.

And

Figure 4. (a) – Melting curves averaged over Hepa1, MEF and RAW264.7 cell lines of bound Keap1 and Keap1 in 3LL cell lysate, and reconstructed curve of free Keap1.

Reviewer 4 Report

This manuscript deals with an interesting topic. It consists of a new theoretical methodology to estimate the concentration of Nrf2 (or another) in the cytoplasm and nucleus. However, it is based on several assumptions that are not entirely clear from the manuscript. For this reason, I consider it important to address the following points.

Major problems:

1. Methodology

a. It is not clear how they obtain the equations

b. Although there is a section for the determination of the volume of the cytoplasm and nucleus, it is not clear how they do it.

c. It is required to include a diagram that clarifies the methodology described.

2. Discussion

a. What advantages would the proposed methodology have compared to the other methodologies that are usually used to determine the activity of Nrf2?:

i. Expression in cytoplasm and nucleus by WB 

ii. Immunocytochemistry

iii. RT-PCR

IV. Expression and activity of the proteins that transcribe Nrf2

v. Others

b. Compare the equipment required in this procedure against that used conventionally (item a); include advantages and disadvantages.

c. There are compounds that modulate the activity of Nrf2 at different levels and forms. Would the proposed methodology be useful in these cases?

Minor issues

• Include equation references in the manuscript such as the one found on line 120.

• Activation of Nrf2 is not exclusive to oxidative stress; please add this information in the manuscript

Author Response

Major problems:

  1. Methodology

“It is not clear how they obtain the equations.”

Our Reply:

For a better understanding of the derivation of the obtained equations, we have included systems of equations (5) and (14), the solution of which leads to final equations. In addition, for clarity, we have included intermediate expressions (22) and (24).

“Although there is a section for the determination of the volume of the cytoplasm and nucleus, it is not clear how they do it.”

Our Reply:

Thank you, we've corrected section “Estimation of cell volumes” of the Materials and Methods to make it clearer how it was done.

The following phrases have been removed:

To measure sizes of cells and their nuclei, the cells were detached from the substrate with a 0.25% trypsin solution, the medium was replaced with the Versene solution, and the cells were placed into a hemocytometer counting chamber.
And

To calculate the volume, V, according to the transmitted light channel, the cells and their nuclei were approximated by a rotation ellipsoid, i.e., the calculation was carried out according to the formula: V = p∙L∙W2/6.

And

The obtained volumes of cells and their nuclei were averaged over 60–80 cells. Dividing cells were disregarded.

New section “Estimation of cell volumes”:

The characteristic sizes of the cells and their nuclei were measured with Leica TCS SP2 confocal microscope (Leica, Germany) in a transmitted light channel (air lens with the magnification ×63 and NA 0.7 was used). For this purpose the cells were detached from the substrate with a 0.25% trypsin solution, the medium was replaced with the Versene solution, and the cells were placed into a hemocytometer counting chamber which is then placed in the microscope holder.  The length, L, and width, W, of the cells and their nuclei were measured. To calculate the volume, V, the cells and their nuclei were approximated by a rotation ellipsoid, i.e., the calculation was carried out according to the formula:  

V = p∙L∙W2/6

(1).

Volumes of cells and their nuclei obtained by expression (1) were averaged over 60–80 cells. Dividing cells were disregarded. The height was considered the same as the width. To verify this assumption, 3LL cells were stained with the CellMaskTM Green plasma membrane stain (Thermo Scientific, USA). Excitation was performed by the 532 nm laser, and fluorescence was recorded in the range 550–600 nm. Three-dimensional images of cells were obtained on the confocal microscope and their length, width and height were measured. We have shown that the height of the cells is the same as their width. The volume of the cytoplasm was considered as the difference between the volume of the cell and the volume of the nucleus, where the latter were determined by the equation (1).

The correctness of determination of the size of the nuclei according to the transmitted light channel was verified for Hepa 1-6 cell line as follows. Nuclei were stained with the SYBR Green fluorescent dye (488 nm excitation, 500–550 nm fluorescence). The average volumes of the nuclei were determined from the data obtained from the transmitted light channel and the fluorescent channel 500–550 nm.

The correctness of the determination of the cells sizes according to microscopy was verified for 3LL line cells using the dynamic light scattering method on a ZetaPALS device (Brookhaven, USA). To do this, one can find the diameter of the sphere, which has the same volume as calculated by expression (1) according to microscopy data and compare it with the hydrodynamic diameter determined from the data of dynamic light scattering. The dynamic light scattering measurement was carried out at the angle of 90° and the accumulation time of 30 seconds in seven replications.

“It is required to include a diagram that clarifies the methodology described.”

Our Reply:

Thanks for the suggestion. We have included a scheme of our approach in Results:

Figure 1. Scheme of our approach to evaluate the effective cytosolic concentration of bioactive agents interacting with a selected intracellular target protein.

Phrases inserted into Results:

To assess the effective cytoplasmic concentration of a bioactive agent, we proposed the following scheme (Figure 1). Cell concentration was determined using flow cytometry. The volumes of the cells and their nuclei were measured using microscopy. For Hepa 1-6 cell line, the average nucleus volumes according to the transmitted light channel and the fluorescent channel were 410 ± 20 and 444 ± 19 mm3, respectively. Thus, according to the data of the transmitted light channel, it is possible to correctly determine the volumes of the nuclei. The cell size determined by the dynamic light scattering is in good agreement with the size measured by the microscopy. For example, the average diameter of the 3LL line cells determined by the microscopy was 13.9 ± 0.4 mm, and the diameter determined by the dynamic light scattering was 13.2 ± 2.7 mm. The volume of the cytoplasm was taken as the difference between the volume of the cell and its nucleus. Knowing the concentration of cells and the volume of the cytoplasm, it is possible to calculate the dilution factor of the studied molecule during cell lysis.

The approach proposed by us was applied for a quantitative description of the Nrf2 system. To estimate the concentration of a molecule in our approach, it is necessary to verify the correctness of the proposed mathematical description of the interactions between the studied molecules.

“Discussion

  1. What advantages would the proposed methodology have compared to the other methodologies that are usually used to determine the activity of Nrf2?:
  2. Expression in cytoplasm and nucleus by WB 
  3. Immunocytochemistry

iii. RT-PCR

  1. Expression and activity of the proteins that transcribe Nrf2
  2. Others
  3. Compare the equipment required in this procedure against that used conventionally (item a); include advantages and disadvantages.”

Our Reply:

Thanks for your question and suggestions.  The work was primarily devoted not to studying the activation of the Nrf2 system, but to measuring the concentrations of the corresponding molecules. As regards the activation of the Nrf2 system, only its very first stages have been studied. Namely, the release of Nrf2 from the complex with Keap1 in response to external action and an increase in the concentration of Nrf2 due to the difference in the rates of degradation of free Nrf2 and Nrf2 in the complex with Keap1 were evaluated. All other stages of the process of activation of the Nrf2 system, such as the entry of Nrf2 into the cell nucleus, the activation of the corresponding genes, and the production of protective cell proteins are not considered in this approach. Therefore, the proposed approach does not cancel a large number of methods used to study the activation of the Nrf2 system, but only complements them in terms of Nrf2 and Keap1 concentrations. In the text of the manuscript, we added an explanation that earlier in the literature, using the Western blot and the corresponding calibration dependences, the amounts of Nrf2 and Keap1 in some cell lines were estimated. It is these data that we used to verify our proposed approach. However, in contrast to the conventional Western blot with calibrations, our approach results in an effective concentration of molecules, i.e., the concentration of only those molecules that have overcome all intracellular barriers and are able to bind to the target protein. In this case, the Nrf2 system can be considered more as a system for testing the effectiveness of various drug delivery systems than on its own. All the equipment used in our approach was also listed and possible alternatives for it were indicated.

Phrases inserted into Discussion:

In the proposed approach aimed at estimation of the concentration of selected molecules, it is necessary, first of all, to determine the concentration of cells and the sizes of cells and their nuclei, which make it possible to calculate the dilution factor of molecules during cell lysis. Then one need to determine the proportion of target protein molecules bound to the selected molecules in the cell lysate, followed by determination of the concentration of the molecules in the cell lysate, based on a simple mathematical description of the interaction between the molecule and the target protein, and subsequent calculation of the concentration of the molecules in the cytoplasm using the dilution factor (Figure 1). Cell concentration can be measured using a hemocytometer counting chamber, as is usually done in many laboratories, but we used flow cytometry as a more accurate method. The characteristic dimensions of cells can be measured by various methods convenient for a particular laboratory. We measured the size of cells detached from the substrate using optical microscopy. The obtained dimensions determined by optical microscopy and dynamic light scattering are in good agreement with each other. Optical microscopy allows one to determine the size of not only the cells, but also their nuclei. However, the size of the nuclei should only be determined if the chosen molecule is known to reside predominantly in the cytoplasm rather than in the nucleus. If this is not known, then it is sufficient to estimate the average concentration of the molecule in the cell.

The main advantage of the proposed approach is that it can be used to estimate not just the concentration of selected molecules, but their effective concentration, i.e. the concentration of only those molecules that are able to interact with the selected target protein. In a cell, the effective concentration of molecules can differ markedly from their average concentration due to, for example, the fact that some of the selected molecules can undergo modifications that block binding to the target protein, or due to the fact that the molecule and the target protein can be located in different cellular compartments. Evaluation of the effective concentration is achieved due to the fact that the behavior of the target protein, rather than the molecule itself, is studied. In other words, the proportion of target protein molecules bound to the studied molecules is estimated, and the effective concentrations of these molecules are calculated using a mathematical interaction model. The model can take into account whether this is a simple two-component interaction or competitive binding with some third component. The proportion of target protein molecules bound to the studied molecules was determined in the proposed approach using CETSA. In world practice, this method is quite common for studying the interaction of proteins with their ligands [22, 40], and we have shown that it can also be used to determine the desired proportion of the target protein.

The proposed approach was tested on the Nrf2 system. On the one hand, this system is interesting in itself, due to its responsibility for protecting cells from oxidative stress and detoxification. On the other hand, it has already been studied in many respects, which makes it possible to use known data to verify the proposed approach. Then, considering Keap1 as the target protein, one can determine the concentration of Nrf2, and if one chooses Nrf2 as the target protein, one can determine the concentration of Keap1. First of all, it was necessary to determine the mathematical model of interactions in this system.

And

“In this work, the Nrf2 system of the cell was considered as a test system on which the proposed approach was checked. However, the approach itself can be very useful for further study of the activation conditions for this system. To determine the required concentrations of molecules, we considered only the first stages of the Nrf2 system activation. Namely, the release of Nrf2 from the complex with Keap1 in response to external action and an increase in the concentration of Nrf2 due to the difference in the rates of degradation of free Nrf2 and Nrf2 in the complex with Keap1. All other stages of the process of activation of the Nrf2 system, such as the entry of Nrf2 into the cell nucleus, the activation of the corresponding genes, and the production of protective cell proteins are not considered in this approach. Therefore, the proposed approach does not cancel a large number of methods used to study the activation of the Nrf2 system, but only complements them in terms of Nrf2 and Keap1 concentrations.”

And

“As a rule, only changes in the relative amounts of Nrf2 and Keap1 in the cell are studied using Western blot, while there are few studies that determine the absolute amounts of these proteins. In the latter case, as a rule, calibration dependencies with known amounts of the recombinant protein are used. This is how the quantities of Nrf2 and Keap1 per cell were obtained [33], which were used by us to verify the proposed approach. Indeed, in all cases, it turned out that the concentrations of Nrf2 and Keap1, estimated using our approach, are in good agreement with these literature data.

The proposed approach makes it possible not only to study the onset of activation of the Nrf2 system, but also to use this system to compare the effectiveness of different drug delivery systems. Indeed, let us consider an arbitrary delivery system for an effector molecule capable of interacting with Keap1 and leading to the release of Nrf2. A number of approaches, such as a Western blot with appropriate protein calibrations, allow estimation of the concentration of this delivery system in the cytoplasm. However, in a number of cases, the delivery system enters the cell, but not the cytosol, being trapped in closed membrane formations - endosomes, with only some proportion entering finally the cytosol. In other words, the concentration of the delivery system in the cytoplasm may differ significantly from the concentration of this system in the cytosol. As already mentioned, the approach proposed by us makes it possible to determine not just the average concentration of a molecule, but its effective concentration, i.e. the concentration of only those molecules that are able to interact with the target protein. Taking into account that sometimes the effector molecule can be cleaved from the delivery system in the cell, in our example this will mean that only the concentration of effector molecules released from endosomes into the cytosol and retaining the ability to bind to Keap1 will be determined. Thus, for the chosen delivery system, it is possible to evaluate its effectiveness in delivering the effector molecule directly to the target protein, in the case under consideration, to Keap1. This, in turn, makes it possible to study the influence of the structure of the delivery system on the efficiency of delivery, and in the long run to significantly improve this efficiency. In the proposed approach, we have described a mathematical model of both a simple two-component interaction (equation (7)) and a triple, competitive interaction (equation (15)). Experimentally, in the first case, cell lysis is performed first, and then CETSA and Western blot with antibodies to Keap1, in the second case, first CETSA, and then cell lysis and Western blot with antibodies to Nrf2.”

The following phrases have been removed:

“The proposed approach makes it possible to analyze both two-component systems, where the concentration of the molecule interacting with the target protein is determined by expression (4), and three-component competitive systems, where the concentration of the molecule interacting with the target protein is determined by expression (9). In the latter case, for the Nrf2 system, expression (13) (special case of expression (9)) is better represented as expression (22). The interaction of a protein molecule containing a monobody to Keap1 can be considered as a binary interaction or ternary competition systems. In the first case, it is possible to obtain melting curves of the Keap1 protein using a Western blot, and in the second case, melting curves of the Nrf2 protein displaced during the interaction.”

“There are compounds that modulate the activity of Nrf2 at different levels and forms. Would the proposed methodology be useful in these cases?”

Our Reply:

Thanks for your question. The concentrations of Nrf2 and Keap1 estimated using our approach do not depend on the cause triggering the release of Nrf2 from the complex with Keap1. This can be either the competition of a certain molecule and Nrf2 for binding to Keap1, as this is considered in our work, or, for example, the release of Nrf2 from the complex with Keap1 due to the modification of Keap1 sensor groups. In any case, the developed approach makes it possible to evaluate how the concentrations of Nrf2 and Keap1 change under some influence on the cell, and thus, to better understand the conditions for the activation of the Nrf2 system.

Phrases inserted into Discussion:

Moreover, the assessment of these concentrations does not depend on the cause underlying the release of Nrf2 from the complex with Keap1. This can be either the competition of a certain molecule and Nrf2 for binding to Keap1, as this is considered in our work, or, for example, the release of Nrf2 from the complex with Keap1 due to the modification of Keap1 sensor groups. In any case, the developed approach makes it possible to evaluate how the concentrations of Nrf2 and Keap1 change under some influence on the cell, and thus, to better understand the conditions for the activation of the Nrf2 system.

“Minor issues

  • Include equation references in the manuscript such as the one found on line 120”

Our Reply:

Thank you. We have inserted all the necessary references to the equations.

“• Activation of Nrf2 is not exclusive to oxidative stress; please add this information in the manuscript”

Our Reply:

Thank you for noticing this. We inserted the necessary information in the Introduction:

In addition to protecting against oxidative stress, Nrf2 serves to protect against environmental pollutants, bacterial and viral toxins, and also has an anti-inflammatory effects [13].”

Round 2

Reviewer 3 Report

The authors' responses to the requested revisions are satisfactory and acceptance for publication is recommended.

Author Response

Thank you very much.

Reviewer 4 Report

An Approach to Evaluate the Effective Cytosolic Concentration of Bioactive Agents Interacting with a Selected Intracellular Target Protein

This manuscript is about a mathematical model to determine the concentration of Nrf2, the authors mention that using this method its effective concentration could be determined.

Abstract:

• Since the proposed model does not apply to bioactive compounds, such as antioxidants, that modulate the expression of Nrf2 or Keap 1, the abstract should be modified.

• I suggest that the authors add the idea described in lines 980-988 in the abstract.

Material and methods

• Although the manuscript is in preparation, it is important to describe briefly and in a general way what fmKeap1 consisted (lines 81-82).

• Verify that equation 1 is to determine the volume of rotation ellipsoid or for a volume sphere

Results:

• Explain the molecular weights of Nrf2  (85, 64 and 28 kDa vs. 100 KDa (other manuscripts))

• The image cuts the idea of line 413.

• How do they activate Nrf2 (lines 523, 555)?

• It would be convenient for them to carry out some statistical method to determine if there is a significant difference between the method described by Cetsa and that of WB (Table 2).

Discussion

• Would this model be helpful to determine Nrf2 concentrations under conditions of oxidative stress induced by external agents (medications, toxins, etc.)?

• It is mentioned that the main of the work is to determine the concentration of a molecule; for this it is necessary to determine its concentration as a first step... (lines 618 and 619). I suggest improving the wording, as this is confusing.

• It is necessary to justify, based on the evidence, why the authors decided to take the MW of Nrf2 as 64 kDa since the MW is usually 100 KDa.

• Considering that this model does not fit the use of compounds that induce and activate Nrf2, I consider that the paragraph of lines 727 and 728 should be modified.

Author Response

“This manuscript is about a mathematical model to determine the concentration of Nrf2, the authors mention that using this method its effective concentration could be determined.

Abstract:

  • Since the proposed model does not apply to bioactive compounds, such as antioxidants, that modulate the expression of Nrf2 or Keap 1, the abstract should be modified.
  • I suggest that the authors add the idea described in lines 980-988 in the abstract.”

Our Reply:

Thanks for your suggestions.

New Abstract:

“To compare the effectiveness of various bioactive agents reversibly acting within a cell on a target intracellular macromolecule, it is necessary to assess effective cytoplasmic concentrations of the delivered bioactive agents. In this work, based on a simple equilibrium model and the Cellular Thermal Shift Assay (CETSA), an approach is proposed to assess effective concentrations of both a delivered bioactive agent and a target protein. This approach was tested by evaluating the average concentrations of nuclear factor erythroid 2-related factor 2 (Nrf2) and Kelch-like ECH-associated-protein 1 (Keap1) proteins in the cytoplasm for five different cell lines (Hepa1, MEF, RAW264.7, 3LL and AML12) and comparing the results with known literature data. The proposed approach makes it possible to analyze both binary interactions and ternary competition systems thus it can have a wide application for the analysis of protein-protein or molecule-protein interactions in the cell. The concentrations of Nrf2 and Keap1 in the cell can be useful not only in analyzing the conditions for the activation of the Nrf2 system, but also for comparing the effectiveness of various drug delivery systems, where the delivered molecule is able to interact with Keap1.”

We removed the old Abstract:

 “To compare the effectiveness of various bioactive agents reversibly acting within a cell on a target intracellular macromolecule, it is necessary to assess cytosolic concentrations of the delivered bioactive agents. In this work, based on a simple equilibrium model and the Cellular Thermal Shift Assay (CETSA), an approach is proposed to assess concentrations of both a delivered bioactive agent and a target protein. This approach is described using the example of molecules capable of interacting with Keap1 protein and activating the antioxidant cellular response of nuclear factor erythroid 2-related factor 2 (Nrf2) system in cells. Using the proposed approach, the average concentrations of Nrf2 and Kelch-like ECH-associated-protein 1 (Keap1) proteins in the cytoplasm were determined for five different cell lines (Hepa1, MEF, RAW264.7, 3LL and AML12). It turned out that if the Nrf2-dependent pathway was not initially activated, the concentrations of Nrf2 or Keap1 were the same in different cell types. The proposed approach makes it possible to analyze both binary interactions and ternary competition systems thus it can has a wide practical application.”

Phrases inserted into Introduction:

“Moreover, knowledge of the concentrations of the components is necessary not only in the quantitative analysis of the activation conditions of the Nrf2 system, but also in the general case when studying any intracellular protein-protein or molecule-protein interactions. In addition, one usually needs not just the average concentration of molecules in the cytoplasm, but its effective concentration. In other words, the concentration of only those molecules that are capable of interacting with the target protein, i.e., at least located with the target protein in the same cellular compartment, is usually required. In this article, based on the Cellular Thermal Shift Assay (CETSA) [20, 21, 22], we proposed a new approach to estimate the effective concentration of a molecule capable of interacting with a selected target protein, both in the case of binary and ternary competitive systems. This approach was tested by evaluating the concentrations of Nrf2 and Keap1 for several cell lines and comparing the results with known literature data.”

The following phrases have been removed from Introduction:

In the article, we proposed a new approach to estimate both the intracellular concentrations of Nrf2 and Keap1 and the Nrf2 competitor molecule for binding to Keap1, based on the Cellular Thermal Shift Assay (CETSA) [20, 21], a simple equilibrium model of interactions, and known interaction affinities between components. This approach can be applied to different binary and ternary competitive systems.

“Material and methods

  • Although the manuscript is in preparation, it is important to describe briefly and in a general way what fmKeap1 consisted (lines 81-82).”

Our Reply:

Thanks for your suggestions.

We inserted an additional phrase in Materials and Methods:

In addition to the R1 monobody, this construct contains a ligand module capable of interacting with EGFR receptors on the cell surface, an endosomolytic module that allows it to exit endosomes, and a carrier module that combines all modules together and imparts additional solubility to the construct.

“• Verify that equation 1 is to determine the volume of rotation ellipsoid or for a volume sphere”

Our Reply:

Thank you. We have checked expression (1). This is a standard expression for determining the volume of a rotation ellipsoid, only expressed not in terms of the semiaxes of the ellipsoid (radii), but in terms of its full axes (diameters).

“Results:

  • Explain the molecular weights of Nrf2  (85, 64 and 28 kDa vs. 100 KDa (other manuscripts))”

And

“Discussion

  • It is necessary to justify, based on the evidence, why the authors decided to take the MW of Nrf2 as 64 kDa since the MW is usually 100 KDa.”

Our Reply:

Thanks for your suggestions. We chose the 64 kDa band for analysis, because, according to the literature data, the 57–68 kDa band corresponds to the Nrf2 fragment, and its intensity changes upon activation of the Nrf2 system. The band at about 30 kDa was also considered to correspond to the Nrf2 fragment according to the literature data. Among all the cell lines studied by us, the band above 64 kDa is observed only for the MEF cell line. In order to observe the full-length Nrf2 band, the proteasome inhibitor MG-132, which causes a significant increase in the Nrf2 concentration, was added to 3LL cells. An increase in the concentration of Nrf2 results in the appearance of a band at 85 kDa. It is impossible to say exactly whether this band corresponds to a very large Nrf2 fragment or a full-sized Nrf2, because there are many isoforms of Nrf2, which differ markedly in molecular weight. Within the framework of our approach, the use of bands around 28, 64, and 85 kDa gives the same melting curve (Figure S3), which corresponds to the melting curve of the full-sized Nrf2. This is also confirmed by the results obtained by us using the proposed approach. Using our approach, it is possible to obtain the concentration of Keap1 in the cytoplasm from the band for Nrf2. On the other hand, the concentration of Keap1 can be obtained from the literature Western blot data with appropriate calibrations for the concentration of Keap1. For the four cell lines, the concentrations obtained by the two different approaches are the same. Moreover, the Nrf2 concentration in the 3LL cell line was estimated from the Nrf2 melting curve. It also matched well with that calculated from the literature data. This confirms not only our approach, but also the fact that the melting curve used corresponds to full-sized Nrf2.

In the Results, we have replaced the phrase

Moreover, the melting curve can be obtained both from the band of about 64 kDa and from the band of about 28 kDa and, importantly, these curves coincide with each other (Figure S3). When the proteasome inhibitor MG-132 is added to 3LL cells, a band of ~ 85 kDa appeared on the blot, which is also visible on lysates of MEF cells (Figure S2). The melting curves obtained from the ~ 85 kDa band coincide with similar curves obtained from the ~ 64 kDa band (Figure S3).

with

“We chose the 64 kDa band to obtain the Nrf2 melting curve, since among all the cell lines studied by us, the band above 64 kDa is observed only for the MEF cell line and, according to the literature data, the 57-68 kDa band corresponds to the Nrf2 fragment, and its intensity changes upon activation of the Nrf2 system [38, 39, 40]. Another band observed at 28 kDa also seems to correspond to the Nrf2 fragment [41]. Moreover, the melting curves obtained from the 64 and 28 kDa bands coincide with each other (Figure S3). When the proteasome inhibitor MG-132, which causes a significant increase in the Nrf2 concentration, was added to 3LL cells, a band of ~ 85 kDa appeared on the blot, which was also visible on lysates of MEF cells (Figure S2). The melting curves obtained from the ~ 85 kDa band coincide with similar curves obtained from the ~ 64 kDa band (Figure S3).”

And phrase

“It should be noted that all the obtained concentrations of Keap1 are in good agreement with the concentrations of Keap1 calculated on the basis of the literature data in the cytoplasm of cells (Table 2), which confirms the applicability of our proposed approach.”

with

“It should be noted that all the obtained concentrations of Keap1 are in good agreement with the concentrations of Keap1 calculated on the basis of the literature data in the cytoplasm of cells (Table 2), which confirms the applicability of our proposed approach and indicates that the melting curve for the Nrf2 fragment selected for analysis (64 kDa) coincides with the melting curve for the full-sized Nrf2.”

In the Discussion, we have replaced the phrase

“But even with the use of this inhibitor, the band at 95-110 kDa was not observed; however, a band of about 85 kDa, close in molecular weight to the full-sized Nrf2, appeared (Figure S3g).”

with

But even with the use of this inhibitor, the band at 95-110 kDa was not observed; however, an increase in the concentration of Nrf2 results in the appearance of a band of about 85 kDa, close in molecular weight to the full-sized Nrf2 (Figure S3g).”

And the phrase

“Thus, the band of about 64 kDa that we have chosen makes it possible to obtain a melting curve corresponding to the melting curve of the full-size Nrf2.”

 with

“Thus, the band of about 64 kDa that we have chosen makes it possible to obtain a melting curve corresponding to the melting curve of the full-size Nrf2. This is also confirmed by the fact that the Keap1 concentrations obtained from the Nrf2 melting curves using the 64 kDa band are in good agreement with the literature data (Table 2).”

“• The image cuts the idea of line 413.”

Our Reply:

Thank you. We have corrected the position of the Figures.

“• How do they activate Nrf2 (lines 523, 555)?”

Our Reply:

Thanks for the question. In both these cases, we have replaced the phrase

the Nrf2 system is initially not activated

 with

almost all of Nrf2 is in the complex with Keap1

“• It would be convenient for them to carry out some statistical method to determine if there is a significant difference between the method described by Cetsa and that of WB (Table 2).”

Our Reply:

Thanks for your suggestions. The manuscript compares our data with literature data, where only mean values with corresponding standard errors are available. We do not know to what extent the given values follow a normal distribution, so the quantitative assessment of the difference in the means will be statistically unjustified. Therefore, one can only compare whether the standard errors of the corresponding values overlap or not. In our case, these errors overlap, which means that there is no significant difference between the compared values. If one assumes a normal distribution, then according to the t-test, it turns out that p is greater than 0.4, i.e., values do not differ significantly from each other

We have replaced the phrase:

“the concentration of Keap1 in the cytoplasm is the same within the error and averaged 269 ± 4 nM.”

with

“the concentration of Keap1 in the cytoplasm is the same within the standard errors of experiments and averaged 269 ± 4 nM.”

“Discussion

  • Would this model be helpful to determine Nrf2 concentrations under conditions of oxidative stress induced by external agents (medications, toxins, etc.)?”

Our Reply:

Thanks for the question. Yes, of course, the proposed approach makes it possible to determine both the concentration of Nrf2 and the concentration of Keap1 under any external influence, including medications and toxins. This was stated in the Discussion in the following phrases:

“In any case, the developed approach makes it possible to evaluate how the concentrations of Nrf2 and Keap1 change under some influence on the cell, and thus, to better understand the conditions for the activation of the Nrf2 system.”

“• It is mentioned that the main of the work is to determine the concentration of a molecule; for this it is necessary to determine its concentration as a first step... (lines 618 and 619). I suggest improving the wording, as this is confusing.”

Our Reply:

Thanks for your suggestion.

We have replaced the phrase:

“In the proposed approach aimed at estimation of the concentration of selected molecules, it is necessary, first of all, to determine the concentration of cells and the sizes of cells and their nuclei”

with

“In the proposed approach aimed at estimation of the concentration of selected molecules, it is necessary, first of all, to determine the number of cells and the sizes of cells and their nuclei”

“• Considering that this model does not fit the use of compounds that induce and activate Nrf2, I consider that the paragraph of lines 727 and 728 should be modified.”

Our Reply:

Thanks for your suggestion. From the melting curves for Nrf2, one can understand whether it is all in a complex with Keap1 or a noticeable fraction of it is in a free state. The latter indicates that the Nrf2 system in such cells is most likely activated due to intracellular causes. The conclusion is made on the basis of the CETSA only, without using a model description, which has its own limitations.

We have replaced the phrase:

We have shown that the CETSA makes it possible to determine in which cells the Nrf2 system is initially activated and in which it is not.

with

We have shown that the CETSA makes it possible to determine in which cells all Nrf2 is in a complex with Keap1, and in which the proportion of free Nrf2 is significant, i.e. in which cells the Nrf2 system is initially activated and in which it is not.